# Consensus Based Stochastic Optimal Control

**Liyao Lyu** [1]   **Jingrun Chen** [2]

## Abstract

We propose a *gradient-free* deep reinforcement learning algorithm to solve *high-dimensional*, finite-horizon stochastic control problems. Although the recently developed deep reinforcement learning framework has achieved great success in solving these problems, direct estimation of policy gradients from Monte Carlo sampling often suffers from high variance. To address this, we introduce the Momentum Consensus-Based Optimization (M-CBO) and Adaptive Momentum Consensus-Based Optimization (Adam-CBO) frameworks. These methods optimize policies using Monte Carlo estimates of the value function, rather than its gradients. Adjustable Gaussian noise supports efficient exploration, helping the algorithm converge to optimal policies in complex, nonconvex environments. Numerical results confirm the accuracy and scalability of our approach across various problem dimensions and show the potential for extension to mean-field control problems. Theoretically, we prove that M-CBO can converge to the optimal policy under some assumptions.

## 1. Introduction

Stochastic optimal control (SOC) problems (Stengel, 1986; Fleming & Rishel, 2012), along with their mean-field variants, have been extensively studied throughout the twentieth century and have had a wide range of applications in various areas, such as finance (Pham, 2009; Fleming & Stein, 2004; Carmona & Durrleman, 2003; Cousin et al., 2011;

---

[1]Department of Computational Mathematics, Science & Engineering, Michigan State University, MI 48824, USA [2]School of Mathematical Sciences and Suzhou Institute for Advanced Research, University of Science and Technology of China, and Suzhou Big Data & AI Research and Engineering Center, Suzhou 215127, China . Correspondence to: Liyao Lyu <lyuliyao@msu.edu>, Jingrun Chen <jingrunchen@ustc.edu.cn>.

*Proceedings of the $42^{nd}$ International Conference on Machine Learning*, Vancouver, Canada. PMLR 267, 2025. Copyright 2025 by the author(s).

Lachapelle et al., 2016; Cardaliaguet & Lehalle, 2018), economics (Guéant, 2009; Gomes et al., 2015; Guéant et al., 2010; Achdou et al., 2014; 2022), chemistry (Welch et al., 2019; Holdijk et al., 2024), and biology (Lachapelle & Wolfram, 2011; Aurell & Djehiche, 2018; Achdou & Lasry, 2019). Readers seeking an overview of these developments may refer to the recent review (Hu & Laurière, 2024). Traditional methods for solving the SOC problem, such as the finite-volume method (Richardson & Wang, 2006; Wang et al., 2003), the Galerkin method (Beard et al., 1997; Beard, 1998), and the monotone approximation method (Forsyth & Labahn, 2007), aim to solve the corresponding Hamilton-Jacobi-Bellman (HJB) equations. However, these methods struggle to scale in high-dimensional spaces due to the curse of dimensionality, where the computational complexity grows exponentially with the dimension of state and action variables. This limitation hinders their application in large-scale systems where efficiency is critical.

Significant advances have been made in addressing the high-dimensional SOC problem by modeling control strategies using deep neural networks, leveraging their capability to approximate functions in high-dimensional spaces. One prominent approach is the value-based method (Li et al., 2024; Lien et al., 2024; Obando Ceron et al., 2024; Zhang et al., 2024; Mou & Zhu, 2024), such as the deep-backward stochastic differential equation (BSDE) method (E et al., 2017; Han et al., 2018; Nüsken & Richter, 2021; Pham et al., 2021). Based on the Bellman principle, the optimal control can be modeled as a function of the value function and its gradient. Therefore, solving the value function from the BSDE that it satisfies can automatically give the optimal control of the SOC. These methodologies are commonly referred to as *model-based* methods because they need an explicit connection between the optimal control and the value function. This kind of connection usually depends on accurate modeling of the transition kernel between different states. However, modeling the (mean-field) transition kernel for a real-world process in practical applications can be extremely challenging (Lyu & Lei, 2023; Lu et al., 2019).

Recently, *model-free* methods have gained attention in control and reinforcement learning (Agrawal et al., 2024; Chen et al., 2024; Chen & Zhang, 2024; Dai et al., 2024; Hisaki & Ono, 2024; Hong et al., 2024; Hu et al., 2024; Park et al., 2024; Tang et al., 2024), such as Deep Q Networks

(DQN) (Mnih et al., 2015), Proximal Policy Optimization (PPO) (Heess et al., 2017; Schulman et al., 2015b; 2017), Trust Region Policy Optimization (TRPO) (Schulman et al., 2015a), Deep Deterministic Policy Gradient (DDPG) (Silver et al., 2014; Lillicrap, 2015) and Soft Actor-Critic (SAC) (Haarnoja et al., 2018b;a). These approaches address this issue by directly optimizing the policy without explicit transition kernel modeling. Nevertheless, these methods rely on the evaluation of policy gradients (Jia & Zhou, 2022a;b) or depend on the action and state space discretization (Gu et al., 2021; Carmona et al., 2023). The evaluation of policy gradients often has high variance and is computationally intensive (Hua et al., 2024), and the discretization of action and state space reintroduces dimensionality constraints. To address this, current methods like PPO, TRPO, and SAC are constrained to a time-independent problem, i.e., an infinite time horizon problem, which allows the reward gradient to be computed iteratively.

However, our work tackles a more general and challenging setting: finite time horizon and model-free stochastic control. In this regime, the assumptions typically required by gradient-based methods—such as model knowledge or discount-based recursion—are not available. Consequently, current approaches face a trade-off between model fidelity and scalability, motivating the need for a method that can achieve robust performance without gradient estimation and state-action discretization.

In this work, we introduce a novel approach to overcome the limitations of both model-based and model-free reinforcement learning methods by applying the Adam-CBO (Chen et al., 2022) framework to high-dimensional SOC problems. Unlike value-based methods, our approach is entirely model-free, directly optimizing the policy without requiring an explicit formulation of the transition kernel. In addition, it is gradient-free, avoiding the high-variance issue associated with policy gradients, and mesh-free, eliminating the need to discretize state and action spaces. These features allow our method to scale efficiently in high-dimensional environments, making it particularly suited for finite-horizon problems where the optimal control is time-dependent. Contrary to concerns that direct policy optimization may lead to local optima, our method demonstrates superior accuracy in handling nonconvex issues, as evidenced by extensive numerical results.

Beyond numerical validation, our study contributes a rigorous theoretical foundation by providing the convergence analysis for the M-CBO method, a simplified version of Adam-CBO without adaptive timestep. This proof establishes that, under certain assumptions, our algorithm reliably converges to the optimal policy, addressing a crucial gap in reinforcement learning for the SOC problem, where theoretical guarantees are often challenging to obtain.

## 2. Problem Formulation

Consider a control problem over a finite time horizon $t \in [0, T]$ for some $T < \infty$. The state space is denoted by $\mathcal{S} \subset \mathbb{R}^d$, and the action space by $\mathcal{A} \subset \mathbb{R}^m$. An agent governs its state process $\mathbf{x}_t$ through an action process $\boldsymbol{\alpha}_t$ with a transition kernel $p(\mathbf{x}'|t, \mathbf{x}, \boldsymbol{\alpha})$ that describes the evolution from state $\mathbf{x}$ to state $\mathbf{x}'$ under the action $\boldsymbol{\alpha}$ at time $t$. The agent's goal is to minimize the combined terminal cost $g(\mathbf{x}_T)$ and the running cost $f(t, \mathbf{x}, \boldsymbol{\alpha})$ incurred during the process. The total cost function is generally represented by

$$J[\boldsymbol{\alpha}] = \mathbb{E}\left[\int_0^T f(t, \mathbf{x}_t, \boldsymbol{\alpha}_t)\mathrm{d}t + g(\mathbf{x}_T)\right].$$

In this work, we model the policy $\boldsymbol{\alpha}(t, \mathbf{x}; \theta)$ as a fully connected neural network parameterized by $\theta$. The rest of the paper will focus on finding the optimal $\theta \in \mathbb{R}^D$ such that it minimizes the cost function $\mathcal{J}(\theta) = J[\boldsymbol{\alpha}[(t, \mathbf{x}; \theta)]$.

## 3. Gradient-free Policy Update

We propose two algorithms to find the optimal policy: M-CBO and Adam-CBO. The Adam-CBO algorithm improves on M-CBO by adaptively adjusting the timestep, resulting in better numerical performance.

### 3.1. Momentum Consensus-Based Optimization

In M-CBO, we begin by initializing a population of $N$ agents represented by $(\boldsymbol{\Theta}, \boldsymbol{\Omega}) = (\Theta^1, \Omega^1, \cdots, \Theta^N, \Omega^N) \in \mathbb{R}^{2ND}$. Here $\Theta^i \in \mathbb{R}^D$ denotes the policy parameterization of the $i$-th agent, and $\Omega^i \in \mathbb{R}^D$ represents its momentum. To exploit the current group of policies, we estimate a consensus policy as

$$\mathcal{M}_\beta(\boldsymbol{\Theta}) = \sum_{i=1}^N \frac{\Theta^i w_\beta(\Theta^i)}{\sum_{j=1}^N w_\beta(\Theta^j)},$$

where $w_\beta(\Theta) = \exp(-\beta \mathcal{J}(\Theta))$. Here $\beta \geq 0$ is an inverse temperature parameter, controlling how strongly each agent's performance (determined by the objective function $\mathcal{J}(\Theta)$) influences the consensus. Using the consensus policy, we define the following dynamics to guide each policy toward consensus:

$$\begin{aligned}
\mathrm{d}\Theta_t^i =& \Omega_t^i \mathrm{d}t - \gamma_1\left(\Theta_t^i - \mathcal{M}_\beta(\boldsymbol{\Theta})\right) + \sigma(t)\mathrm{d}W_{\theta,t}^i, \\
\mathrm{d}\Omega_t^i =& -m\left(\Theta_t^i - \mathcal{M}_\beta(\boldsymbol{\Theta})\right)\mathrm{d}t \\
& - \gamma_2 \Omega_t^i \mathrm{d}t + \sqrt{m}\sigma(t)\mathrm{d}W_{\omega,t}^i,
\end{aligned} \quad (1)$$

where $m$, $\gamma_1$, and $\gamma_2$ are positive constants and $W_{\theta,t}^i$, $W_{\omega,t}^i$ are $D$ dimensional Wiener processes that introduce stochasticity into the dynamics. This facilitates the exploration of unknown regions, with a parameter $\sigma(t)$ regulating the exploration strength. Using the Euler-Maruyama (EM) scheme

**Algorithm 1** Consensus Based Optimization with Momentum

> **Input:** time step $\lambda$, Number of player $N$, Batch size $M$, total time $t_N$, parameters $\beta, \gamma_1, \gamma_2, m$
> Initialize $\Theta_0^i \sim \mathcal{N}(0, \mathbb{I}_D)$, $i = 1, \ldots, N$
> Initialize $\Omega_0^i = 0$, $i = 1, \ldots, N$;
> **for** $t = 0$ **to** $t_N$ **do**
>   Partition the indices $\{1, 2, \ldots, N\}$ into batches $B^1, \ldots, B^{\frac{N}{M}}$, each containing $M$ particles
>   **for** $j = 1$ **to** $\frac{N}{M}$ **do**
>     $\mathcal{J}^i = \mathcal{J}(\Theta_t^i)$, where $i \in B^j$
>     $M = \sum\limits_{k \in B^j} \frac{\Theta_t^k w^k}{\sum\limits_{i \in B^j} w^i}$, where $w^i = \exp\left(-\beta \mathcal{J}^i\right)$
>     Update the policies and their momentum:
>
> $$\Theta_{t+1}^i = \Theta_t^i + \lambda \Omega_t^i - \gamma_1 \lambda (\Theta_t^i - M) + \sqrt{\lambda} \xi_\theta^i,$$
> $$\Omega_{t+1}^i = \Omega_t^i - \lambda m (\Theta_t^i - M) - \lambda \gamma_2 \Omega_t^i + \sigma(t) \sqrt{\lambda m} \xi_\omega^i,$$
>
>     where $\xi_\theta^i, \xi_\omega^i \sim \mathcal{N}(0, \mathbb{I}_D)$
>   **end for**
> **end for**
> **Output:** $\Theta_{t_N}^i$, $i = 1, \ldots, N$

for Equation (1), we get the M-CBO algorithm, as detailed in Algorithm 1.

The original CBO method (Fornasier et al., 2024) aims to achieve a monotonic reduction in the distance between the optimal policy $\tilde{\theta}$ and the policies of agents. Specifically, this is represented as: $\frac{1}{N} \sum_{i=1}^N \|\Theta_t^i - \tilde{\theta}\|^2 \simeq \int \|\theta - \tilde{\theta}\|^2 \mathrm{d}\mu_t(\theta)$, where $\mu_t$ represents the law of agents $\Theta_t$. Our method minimizes a combined expression $\frac{1}{N} \sum_{i=1}^N \left( \|\Theta_t^i - \tilde{\theta}\|^2 + m^{-1} \|\Omega_t^i\| \right) \simeq \int \|\theta - \tilde{\theta}\|^2 + m^{-1} \|\omega\|^2 \mathrm{d}\rho_t(\theta, \omega)$, where $\rho_t$ represents the joint distribution of policies $\Theta$ and $\Omega$ at time $t$. In particular, the M-CBO method does not force the monotonic reduction of $\frac{1}{N} \sum_{i=1}^N \|\Theta_t^i - \tilde{\theta}\|^2$, allowing for the additional momentum term $\omega$ to enhance the exploration capability. It provides greater flexibility and reduces the risk of becoming trapped in local minima; see Section 4 for a more detailed analysis.

### 3.2. Adaptive Momentum Consensus-Based Optimization

In the Adam-CBO method, we extend M-CBO by replacing the constant momentum term $m$ with an adaptive term based on the inverse of the second moment of the agents' policies. Specifically, we replace $m$ with $(\mathcal{V}_\beta[\Theta] + \epsilon \mathbf{I})^{-1}$, where $V_\beta[\Theta]$ is the second moment defined as:

$$V_\beta[\Theta] = \sum_{i=1}^N \frac{(\Theta^i - \mathcal{M}_\beta[\Theta])^2 w_\beta(\Theta^i)}{\sum_{j=1}^N w_\beta(\Theta^j)}.$$

**Algorithm 2** Consensus-based Optimization with Adaptive Momentum

> **Input:** time step $\lambda$, Number of player $N$, Batch size $M$, total time $t_N$, parameters $\beta, \beta_1, \beta_2$
> Initialize $\Theta_0^i \sim \mathcal{N}(0, \mathbb{I}_D)$, $i = 1, \ldots, N$
> Initialize $\Omega_0^i = 0$, $i = 1, \ldots, N$
> Initialize $M_0, V_0 = 0$
> **for** $t = 0$ **to** $t_N$ **do**
>   Partition the indices $\{1, 2, \ldots, N\}$ into batches $B^1, \ldots, B^{\frac{N}{M}}$, each containing $M$ particles
>   **for** $j = 1$ **to** $\frac{N}{M}$ **do**
>     $\mathcal{J}^i := \mathcal{J}(\Theta_t^i)$, where $i \in B^j$
>     $M = \sum\limits_{k \in B^j} \frac{\Theta_t^k w^k}{\sum\limits_{i \in B^j} w^i}$
>     $V = \sum\limits_{k \in B^j} \frac{(\Theta_t^k - M)^2 w^k}{\sum\limits_{i \in B^j} w^i}$
>     Update the moving average moment estimate:
>
> $$M_{t+1} = \beta_1 M_t + (1 - \beta_1) M, \quad \hat{M}_{t+1} = \frac{M_{t+1}}{1 - \beta_1^t},$$
> $$V_{t+1} = \beta_2 V_t + (1 - \beta_2) V, \quad \hat{V}_{t+1} = \frac{V_{t+1}}{1 - \beta_2^t}$$
>
>     Update the policies and their momentum:
>
> $$\Theta_{t+1}^i = \Theta_t^i + \lambda V_t^i,$$
> $$\Omega_{t+1}^i = \Omega_t^i - \lambda \operatorname{Diag}(\hat{V}_t^i + \epsilon)^{-1} (\Theta_t^i - \hat{M}_t)$$
> $$+ \gamma \lambda \Omega_t^i + \sigma(t) \sqrt{\lambda} \xi_i,$$
>
>     where $\xi_i \sim \mathcal{N}(0, \mathbb{I}_D)$
>   **end for**
> **end for**
> **Output:** $\Theta_{t_N}^i$, $i = 1, \ldots, N$

In particular, $\epsilon = 1 \times 10^{-8}$ is used to keep the positivity. This adaptive adjustment introduces a mechanism similar to the Adam optimizer, where updates are scaled by a normalized second moment, allowing for faster convergence and improved numerical performance. The detailed algorithm is shown in Algorithm 2.

## 4. Convergence Analysis

In Section 3, we propose two dynamics that converge to the consensus policies. A natural question we want to answer here is whether policies can converge to the optimal policies. From the theoretical perspective, for simplicity, we focus on proving the convergence of the M-CBO method in this work. We begin by establishing the well-posedness of the M-CBO method, ensuring the uniqueness and existence of solutions under certain regularity conditions on the cost function $\mathcal{J}$.

**Assumption 4.1.** The following assumptions are imposed

on the cost function $\mathcal{J}$

1. There exist $\tilde{\theta}$ such that $\mathcal{J}(\tilde{\theta}) = \inf_{\theta} \mathcal{J}(\theta) =: \underline{J}$. Also, it is bounded from above by $\sup \mathcal{J} \leq \bar{J}$.

2. The cost function $\mathcal{J}$ is locally Lipschitz continuous $\|\mathcal{J}[\theta_1] - \mathcal{J}[\theta_2]\| \leq L_J(\|\theta_1\| + \|\theta_2\|)\|\theta_1 - \theta_2\|$.

3. There exists a constant $c_{\mathcal{J}} > 0$ such that $\mathcal{J}(\theta) - \underline{J} \leq c_{\mathcal{J}}(1 + \|\theta\|^2)$.

4. There exist $\delta_J, R_0, \eta, \mu > 0$ such that $\|\theta - \tilde{\theta}\| \leq \frac{(\mathcal{J}-\underline{J})^{\mu}}{\eta}$, for all $\theta \in B_{\theta,R_0}(\tilde{\theta}) = \{\theta : \|\theta - \tilde{\theta}\| \leq R_0\}$, and $\mathcal{J}(\theta) - \underline{J} > \delta_J$ for all $\theta \in \left(B_{\theta,R_0}(\tilde{\theta})\right)^c$.

5. The parameters we choose $\sigma(t)$ has upper and lower bound $\underline{\sigma} \leq \sigma(t) \leq \bar{\sigma}$.

**Theorem 4.2.** *Under the Assumption 4.1, for each $N \in \mathbb{N}$, the stochastic differential equation (1) has a unique strong solution $\left\{\left(\boldsymbol{\Theta}_t^{(N)}, \boldsymbol{\Omega}_t^{(N)}\right) | t > 0\right\}$ for any initial condition $\left(\boldsymbol{\Theta}_0^{(N)}, \boldsymbol{\Omega}_0^{(N)}\right)$ satisfying $\mathbb{E}\left(\|\boldsymbol{\Theta}_0^{(N)}\| + \|\boldsymbol{\Omega}_0^{(N)}\|\right) \leq \infty$.*

*Proof.* See Appendix A. $\square$

By letting the number of agents $N \to \infty$ in Equation (1), the mean-field limit of the model is formally given by the following McKean–Vlasov stochastic differential equation

$$
\begin{aligned}
\mathrm{d}\bar{\Theta}_t =& \bar{\Omega}_t \mathrm{d}t - \gamma_1\left(\bar{\Theta}_t - \mathcal{M}_{\beta}[\mu_t]\right)\mathrm{d}t + \sigma(t)\mathrm{d}W_{\theta,t}, \\
\mathrm{d}\bar{\Omega}_t =& -m\left(\bar{\Theta}_t - \mathcal{M}_{\beta}[\mu_t]\right)\mathrm{d}t \\
& -\gamma_2\bar{\Omega}_t\mathrm{d}t + \sqrt{m}\sigma(t)\mathrm{d}W_{\omega,t},
\end{aligned}
\tag{2}
$$

where $\mathcal{M}_{\beta}[\mu] = \frac{\int \theta \exp(-\beta\mathcal{J}(\theta)\mu(\mathrm{d}\theta))}{\int \exp(-\beta\mathcal{J}(\theta)\mu(\mathrm{d}\theta))}$, $\mu_t(\theta) = \int \rho_t(\theta, \mathrm{d}\omega)$, and $\rho_t = Law(\bar{\Theta}_t, \bar{\Omega}_t)$. Then the corresponding Fokker-Planck equation is

$$
\begin{aligned}
\partial_t \rho_t =& -\nabla_{\theta}\cdot\left((\omega - \gamma_1(\theta - \mathcal{M}_{\beta}[\mu_t]))\rho_t\right) \\
& +\nabla_{\omega}\cdot\left((m(\theta - \mathcal{M}_{\beta}[\mu_t]) + \gamma_2\omega)\rho_t\right) \\
& +\frac{\sigma(t)^2 m}{2}\Delta_{\omega}\rho_t + \frac{\sigma(t)^2}{2}\Delta_{\theta}\rho_t.
\end{aligned}
\tag{3}
$$

Next, we will prove the above equation (2) and (3) are well-posed.

**Theorem 4.3.** *Let $\mathcal{J}$ satisfy the Assumption 4.1 and $\rho_0 \in \mathcal{P}_4(\mathbb{R}^D \times \mathbb{R}^D)$. Then there exists a unique nonlinear process $(\bar{\Theta}, \bar{\Omega}) \in \mathcal{C}\left([0,T], \mathbb{R}^D \times \mathbb{R}^D\right), T > 0$, satisfying (2) with initial distribution $(\bar{\Theta}, \bar{\Omega}) \sim \rho_0$ in the strong sense, and $\rho_t = Law(\bar{\Theta}, \bar{\Omega}) \in \mathcal{C}\left([0,T], \mathcal{P}_4(\mathbb{R}^D \times \mathbb{R}^D)\right)$ satisfies the corresponding Fokker-Planck equation (3) in the weak sense with $\lim_{t\to\infty} \rho_t = \rho_0$.*

*Proof.* See Appendix B. $\square$

Then we present the result showing that (2) and (3) model the mean-field limit of Equation (1).

**Theorem 4.4.** *Let $\mathcal{J}$ satisfy Assumption 4.1 and $\rho_0 \in \mathcal{P}_4(\mathbb{R}^D \times \mathbb{R}^D)$. For any $N \geq 2$, assume that $\{(\Theta_t^{(i,N)}, \Omega_t^{(i,N)})_{t\in[0,T]}\}_{i=1}^N$ is the unique solution to the particle system (1) with $\rho_0^{\otimes N}$-distributed initial data $\{(\Theta_0^{(i,N)}, \Omega_0^{(i,N)})\}_{i=1}^N$. Then the limit (denoted by $\rho$) of the sequence of the empirical measure $\rho^N = \frac{1}{N}\sum_{i=1}^N \delta_{\left(\Theta^{(i,N)}, \Omega^{(i,N)}\right)}$ exists. Moreover, $\rho$ is deterministic and it is the unique weak solution to PDE (3).*

*Proof.* See in Appendix C. $\square$

To prove the global convergence of the M-CBO method, we define the energy functional as

$$
E[\rho] = \frac{1}{2}\int \|\theta - \tilde{\theta}\|^2 + m^{-1}\|\omega\|^2 \mathrm{d}\rho.
\tag{4}
$$

The above definition $E[\rho]$ provides a measure of the distance between the distribution of the agents $\rho$ and the Dirac measure at $(\tilde{\theta}, 0)$, denoted as $\delta_{(\tilde{\theta},0)}$. Specifically, we have the relationship $2E[\rho_t] = W_2^2(\rho_t, \delta_{(\tilde{\theta},0)})$.

**Theorem 4.5.** *Let $\mathcal{J}$ satisfy the Assumption 4.1. Moreover, let $\rho_0 \in \mathcal{P}_4(\mathbb{R}^{2D})$ and $(\tilde{\theta}, 0) \in supp(\rho_0)$. By choosing parameters $\sigma(t)$ is exponentially decaying as $\sigma(t) = \sigma_1 \exp(-\sigma_2 t)$ with $\sigma_1 > 0$ and $\sigma_2 > 1$ and $\lambda = \max\{m, \gamma_1\} \geq 2\sigma_2$ and $\gamma = \min\{\gamma_1, \gamma_2\} > 0$. Fix any $\epsilon \in (0, E[\rho_0])$ and $\tau \in (0, 1 - \frac{2\sigma_2}{\lambda})$, and define the time horizon*

$$
T^* := \frac{1}{(1-\tau)\lambda}\log\left(\frac{E[\rho_{T_0}]}{\epsilon}\right)
\tag{5}
$$

*Then there exists $\beta > 0$ such that for all $\beta > \beta_0$, if $\rho \in \mathcal{C}([0,T^*], \mathcal{P}_4(\mathbb{R}^{2D}))$ is a weak solution to the Fokker-Planck equation in the time interval $[0,T^*]$ with initial condition $\rho_0$, we have*

$$
\min_{t\in[0,T^*]} E[\rho_t] \leq \epsilon.
$$

*Furthermore, until $E[\rho_t]$ reaches the prescribed accuracy $\epsilon$, we have the exponential decay*

$$
E[\rho_t] \leq E[\rho_0]\exp(-(1-\tau)\lambda t)
\tag{6}
$$

*and, up to a constant, the same behavior for $W_2^2(\rho_t, \delta_{(\tilde{\theta},0)})$.*

*Proof.* See Appendix D. $\square$

# 5. Numerical Results

We evaluate the performance of the Adam-CBO method across various problem settings, including the linear quadratic control problem in $1, 2, 4, 8,$ and $16$ dimensions, the Ginzburg-Landau model, and the systemic risk mean-field control problem with $50, 100, 200, 400, 800$ agents. Even though our method is model-free, which means it does not depend on the known explicit knowledge of the transition kernel as well as the precise dependency of the value function $u(t, \mathbf{x})$ on the optimal control $\boldsymbol{\alpha}(t, \mathbf{x}, \nabla u, \text{Hess } u)$. The value function is expressed as:

$$u(t, \mathbf{x}) = \inf_{\alpha \in \mathcal{A}} \mathbb{E}\left[\int_t^T f(s, \mathbf{x}_s, \boldsymbol{\alpha}_s)\mathrm{d}s + g(X_T)\,|x(t) = x\right].$$

To measure the accuracy of our method, we compare $u(t, \mathbf{x})$ or the $\|\boldsymbol{\alpha}(t, \mathbf{x}, \nabla u, \text{Hess } u) - \boldsymbol{\alpha}(t, \mathbf{x}; \theta)\|$ as a metric. Our code is available at https://github.com/Lyuliyao/ADAM_CBO_control.

## Linear Quadratic Control Problem

We begin by considering a classical linear quadratic Gaussian (LQG) control problem. The value function is known as $u(t, \mathbf{x}) = -\ln\left(\mathbb{E}\left[\exp\left(-g\left(\mathbf{x} + \sqrt{2}\mathbf{W}_{T-t}\right)\right)\right]\right)$, which we refer to Appendix E for details. The numeric value of $u(t, \mathbf{x})$ can be computed by Monte Carlo (MC) estimation directly as a reference to measure the accuracy.

We investigate the LQG problem in dimension $d = 1, 2, 4, 8,$ and $16$, with a terminal time of $T = 1$ and a timestep of $\frac{T}{20}$. We compare our method with the BSDE method in (Han et al., 2018). In both methods, the number of SDE to compute the value function is 64 and the learning rate is $1 \times 10^{-2}$. In M-CBO and Adam-CBO methods, the number of agents is specified as $N = 5000$, and $M = 50$ agents are randomly selected to update in each step.

The value function $u(t = 0, \mathbf{x} = (0, \ldots, 0))$ for two different terminal costs - a convex cost: $g(\mathbf{x}) = \ln \frac{1 + \|\mathbf{x}\|^2}{2}$ and a double-well terminal cost: $g(\mathbf{x}) = \ln \frac{1 + (\|\mathbf{x}\|^2 - 1)^2}{2}$ is illustrated in Figure 1 across varying dimensions. The value function from MC estimation is worked as a reference. The value function of Adam-CBO and M-CBO methods is computed from the expectation of 5000 controlled dynamics. The value function of the BSDE method is a direct output of the neural network.

In the convex terminal cost, we can see that both the M-CBO method and the Adam-CBO method outperform the BSDE method in a low-dimensional setting. As the dimensionality increases, Adam-CBO continues to outperform the BSDE method, demonstrating its scalability. Consequently, in the remaining examples, we focus exclusively on the Adam-CBO method because of its superior performance in high

dimensions.

In the case of the double-well terminal cost, which is non-convex, our method shows significantly improved accuracy over the BSDE approach. This enhancement can be attributed to several factors. First, CBO-based methods have a higher likelihood of converging to global minima in non-convex settings. Secondly, although both methodologies utilize a discretization of the 20 time steps during the training phase, our approach facilitates additional refinement of the time steps when assessing the cost function, thereby improving precision. In contrast, the structure of the neural network of the BSDE method is inherently tied to the chosen discretization, necessitating the same time step for both training and evaluation, thus limiting flexibility. We

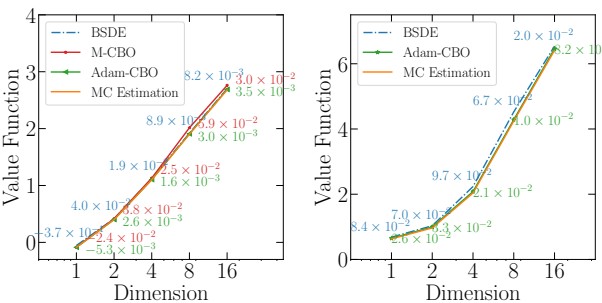

*Figure 1.* The value function $u(t = 0, \mathbf{x} = (0, \ldots, 0))$ evaluated using BSDE method, M-CBO method, Adam-CBO method (*our method*), and MC estimation (*reference*) for problems in $1, 2, 4, 8,$ and $16$ dimensions. (a) The terminal cost function $g(\mathbf{x}) = \ln \frac{1 + \|\mathbf{x}\|^2}{2}$. (b) The terminal cost function $g(\mathbf{x}) = \ln \frac{1 + (\|\mathbf{x}\|^2 - 1)^2}{2}$.

also visualize the function $u(t, x)$ in the one-dimensional case for both types of terminal costs in Figure 2 and Figure 3. It is evident that our method aligns more closely with the exact solution than the BSDE-based method.

We further investigate the influence of batch size (the number of control processes to compute the cost function) on the problem. We consider a $4$ dimensional problem with a nonconvex terminal cost given by $g(\mathbf{x}) = \ln \frac{1 + (\|\mathbf{x}\|^2 - 5)^2}{2}$. Figure 4 illustrates the value function $u(t = 0, 0, 0, 0, 0)$ evaluated under varying batch sizes during training and compared with a precise estimation of the MC that uses a sufficiently large sample size. The first insight is that the accuracy of training is sensitive to batch size in the training process, which inspired us to develop an improved sampling method to enhance the efficiency of the sampling process in the future. Additionally, our method consistently demonstrates greater accuracy than the BSDE-based approach, confirming its robustness in higher-dimensional and nonconvex settings.

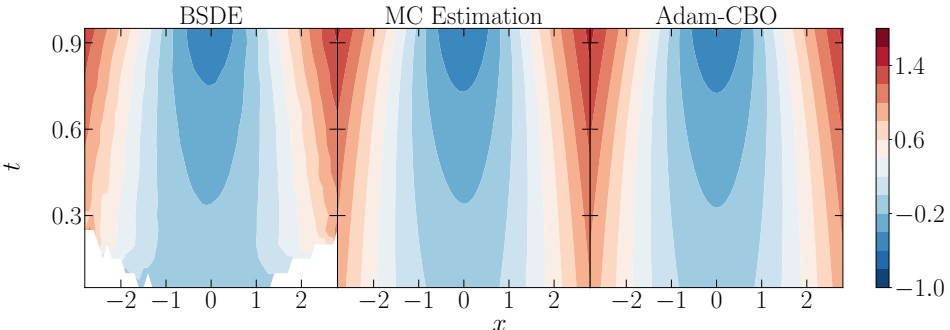

*Figure 2.* The value function $u(t, x)$ in the one-dimensional case, computed using BSDE method, MC Estimation (*reference*), and Adam-CBO (*our method*), with terminal cost $g(\mathbf{x}) = \ln \frac{1+\|\mathbf{x}\|^2}{2}$.

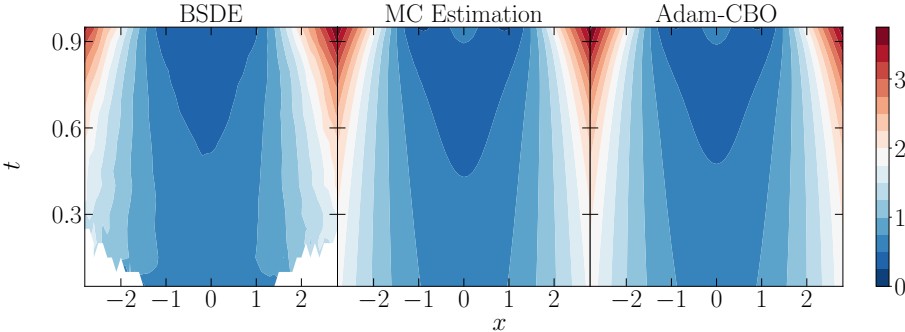

*Figure 3.* The value function $u(t, x)$ in one-dimensional case, computed using BSDE method, MC Estimation (*reference*), and Adam-CBO (*our method*), with terminal cost $g(\mathbf{x}) = \ln \frac{1+(\|\mathbf{x}\|^2-1)^2}{2}$.

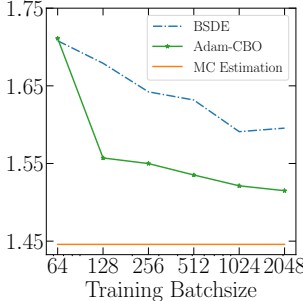

*Figure 4.* The value function $u(t = 0, 0, 0, 0, 0)$ of 4D LQC problem, computed using BSDE method, MC Estimation (*reference*), and Adam-CBO (*our method*), with terminal cost $g(\mathbf{x}) = \ln \frac{1+(\|\mathbf{x}\|^2-5)^2}{2}$, evaluated under varying sample sizes per step.".

### Ginzburg-Landau Model

We also consider the problem of controlling superconductors in an external electromagnetic field, modeled using the stochastic Ginzburg-Landau theory. The dynamics are given by

$$d\mathbf{x}_t = \mathbf{b}(\mathbf{x}_t, \alpha_t)dt + \sqrt{2}d\mathbf{W}_t, \qquad (7)$$

where the drift term is defined as

$$b(\mathbf{x}, a) = -\nabla_{\mathbf{x}} U(\mathbf{x}) + 2\alpha\boldsymbol{\omega}.$$

$U$ here is the Ginzburg-Landau free energy, while $\boldsymbol{\omega} \in \mathbb{R}^d$ specifies the spatial domain of the external field. For further implementation specifics, see Appendix E.

Since this problem lacks an exact analytical solution, we assess the performance of our trained control $\alpha(t, \mathbf{x}; \theta)$ by comparing it to the theoretically optimal control $-\boldsymbol{\omega} \cdot \nabla_{\mathbf{x}} u(t, \mathbf{x},)$, where $u(\mathbf{x}, t)$ is the value function. Notably, this value function is different from the last case with an analytical solution; it was computed by taking the expectation of running controlled dynamics and its gradient is computed by taking the finite difference of two starting states. Therefore, this comparison is not intended as a true error metric. Instead, it serves to evaluate the consistency between our trained control and the theoretically optimal control, which many value-based methods use to define the loss.

We start with a simple case with $d = 2$, $\mu = 10$, $\lambda = 0.2$. We compare the distribution of $x_1$ before and after the control in Figure 5. One can find that before the control the particles will stay in a stable state $-1, 1$, while after control

the particles will stay near 0. Additionally, a comparative

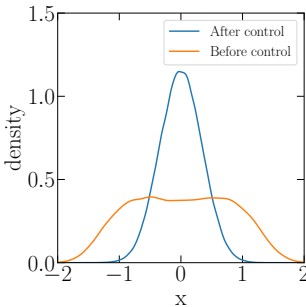

Figure 5. Distribution of $x_1$ before and after control in the 1D Ginzburg-Landau model.

analysis between $\alpha(t, \mathbf{x})$ and $-\boldsymbol{\omega} \cdot \nabla_{\mathbf{x}} u(t, \mathbf{x})$ is conducted, as illustrated in Figure 6. The results demonstrate the consistency between these two functions. We also test our method on $d = 4, 8, 16, 32$. The comparison between $\alpha(t, \mathbf{x}; \theta)$ and $-\boldsymbol{\omega} \cdot \nabla_{\mathbf{x}} u(t, \mathbf{x})$ is shown in Figure 7. Here $(t, \mathbf{x})$ is randomly sampled from 1000 control dynamics.

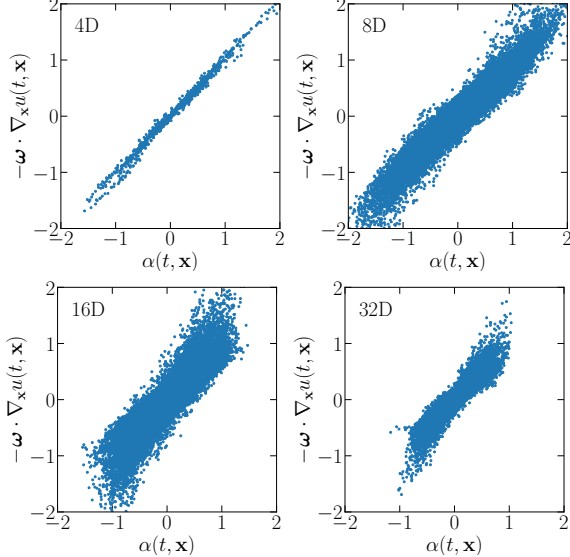

Figure 7. Ginzburg-Landau model with $d = 4, 8, 16, 32$ respectively. The $x$-axis shows the $\alpha(0.5, \mathbf{x})$ and $y$-axis shows the $-\boldsymbol{\omega} \cdot \nabla_{\mathbf{x}} u(0.5, \mathbf{x})$

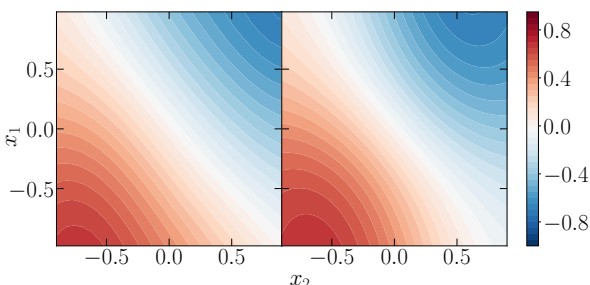

Figure 6. The left figure shows the $\alpha(0.5, \mathbf{x}; \theta)$ and the right figure shows the $-\boldsymbol{\omega} \cdot \nabla_{\mathbf{x}} u(0.5, \mathbf{x})$ computed by our method for the 2D Ginzburg-Landau model.

## Systemic Risk Mean Field Control

In practical applications, there are scenarios where numerous indistinguishable agents, such as multiple traders engaged in buying and selling stocks within financial markets, create a complex, multi-dimensional problem. However, when these traders share similar risk preferences, analyzing the behavior of a single representative trader can suffice to understand the dynamics of the entire group. For example, for a problem with $n$ agents, the control can be modeled as $\boldsymbol{\alpha}^i(\mathbf{x}_i, \mu; \theta)$, where $\mu$ is the empirical measure of the $\{\mathbf{x}_i\}_{i=1}^n$ and $\theta$ are parameters in the neural network. For further details on the network construction used in this setup, refer to Appendix F.

Consider the systemic risk mean field control problem,

detailed in Appendix E. The control policy is initially trained using a delta distribution centered on $x_0$ and $n = 100$ and then tested against different values of $n = 50, 100, 200, 400, 800$. Furthermore, the value function is evaluated by taking the expectation of controlled dynamics starting from different initial distributions $\mu_0$, including Gaussian random variable $x_0 = \mathcal{N}(0, 0.1)$, mixture of two Gaussian random variables $x_0 = p(-k + \theta y) + (1 - P)(k + \theta z)$ with $P$ a Bernoulli random variable with parameter $\frac{1}{2}$, $k = \frac{\sqrt{3}}{10}$, $\theta = 0.1$, $y, z \sim N(0, 1)$ and mixture of three Gaussian random variables: $x_0 = [-k_{\lfloor 3U \rfloor = 0} + k_{\lfloor 3U \rfloor = 1}] + \theta y$ with $k = 0.3$, $\theta = 0.07$, $y \sim N(0, 1)$. The corresponding value functions for each scenario are shown in Table 1, 2, 3, respectively. Our method demonstrates robust generalization across these diverse conditions, in contrast to value-function-based approaches where the control strategy is tied to the specific value function. Since value functions are highly sensitive to initial conditions, traditional methods require retraining for each new initial scenario, limiting their ability to generalize effectively.

## Multi-Agent Robotic Systems

We present numerical results for a multi-agent robotic system consisting of $n$ agents. Each agent $i$ computes an optimal policy $\boldsymbol{\alpha}^i$ to navigate from a predefined starting point to a target while avoiding obstacles in the environment. The original problem is formulated as an open-loop control problem with constraints (Abdul et al., 2024). However, we reformulate it as a feedback control problem by incorporating penalty terms to handle constraints. For further details

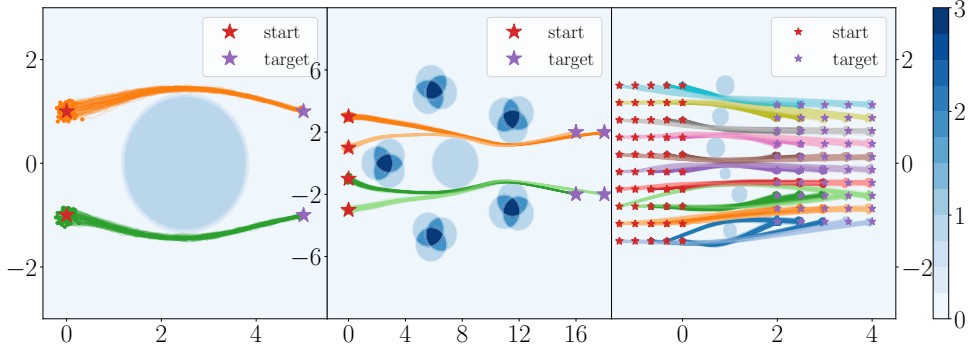

*Figure 8.* Learned trajectories for multi-agent navigation from initial configurations to target destinations while avoiding obstacles. Results are shown for systems with 2, 4, and 50 agents.

*Table 1.* The value function $u(t, \mu)$ evaluated by Adam-CBO method with $\mu = \mathcal{N}(0, 1)$ in the mean filed control problem.

| TIME | $n = 50$ | $n = 100$ | $n = 200$ | $n = 400$ | EXACT |
|------|----------|-----------|-----------|-----------|-------|
| 0.0 | 0.607 | 0.614 | 0.618 | 0.619 | 0.616 |
| 0.1 | 0.553 | 0.559 | 0.563 | 0.564 | 0.561 |
| 0.2 | 0.498 | 0.504 | 0.507 | 0.508 | 0.506 |
| 0.3 | 0.442 | 0.447 | 0.450 | 0.451 | 0.449 |
| 0.4 | 0.384 | 0.388 | 0.391 | 0.391 | 0.390 |
| 0.5 | 0.323 | 0.326 | 0.329 | 0.329 | 0.329 |
| 0.6 | 0.258 | 0.260 | 0.262 | 0.263 | 0.262 |
| 0.7 | 0.187 | 0.188 | 0.190 | 0.190 | 0.190 |
| 0.8 | 0.106 | 0.107 | 0.108 | 0.108 | 0.108 |
| 0.9 | 0.010 | 0.010 | 0.010 | 0.010 | 0.010 |

*Table 2.* The value function $u(t, \mu)$ with $\mu$ being a mixture of two Gaussian random variables in the mean filed control problem.

| TIME | $n = 50$ | $n = 100$ | $n = 200$ | $n = 400$ | EXACT |
|------|----------|-----------|-----------|-----------|-------|
| 0.0 | 0.621 | 0.628 | 0.633 | 0.634 | 0.630 |
| 0.1 | 0.567 | 0.574 | 0.578 | 0.579 | 0.576 |
| 0.2 | 0.513 | 0.518 | 0.522 | 0.523 | 0.521 |
| 0.3 | 0.457 | 0.462 | 0.465 | 0.466 | 0.465 |
| 0.4 | 0.399 | 0.404 | 0.407 | 0.408 | 0.407 |
| 0.5 | 0.339 | 0.343 | 0.346 | 0.346 | 0.346 |
| 0.6 | 0.276 | 0.279 | 0.281 | 0.281 | 0.281 |
| 0.7 | 0.207 | 0.209 | 0.211 | 0.211 | 0.211 |
| 0.8 | 0.129 | 0.131 | 0.132 | 0.132 | 0.132 |
| 0.9 | 0.039 | 0.040 | 0.040 | 0.040 | 0.040 |

on this transformation, we refer the reader to Appendix E.3. The numerical results, presented in Figure 8, demonstrate the performance of our method for systems with 2, 4, and 50 agents. Figure 8 demonstrates successful trajectory generation between initial and target states for systems with 2, 4, and 50 agents, while ensuring collision-free navigation through obstacle-rich environments. These results validate our method's capability to solve high-dimensional optimal control problems that lack explicit analytical solutions.

**Reinforcement Learning Task**

We also compare our method with DDPG, PPO, SAC, TD3, TQC, and CrossQ (using the stable-baselines3 implement https://github.com/araffin/sbx) on Pendulum-v1 as well as PPO and DQN on CartPole-v1. The numerical results can be found in Figure 9 and the computational cost is shown in Table 4. While Adam-CBO has higher runtime, it converges to the optimal policy much faster in terms of learning efficiency. However, we would like to stress that these results are not directly comparable in a strict sense. Most of the baseline methods optimize multiple components—for example, PPO jointly op-

timizes a policy and a value function, and SAC optimizes two Q-functions and a policy. In contrast, our method optimizes only the policy. If we were to directly replace the gradient-based optimizer within an existing method like PPO with Adam-CBO, we do not expect it to outperform the full method in that specific setup. The main advantage of Adam-CBO lies in its applicability to broader, more general settings, particularly when gradients are unavailable or unreliable.

## 6. Conclusion

In this work, we present a framework for solving high-dimensional stochastic optimal control problems. Compared with the existing method, our method is gradient-free, which eliminates the high variance in the Monte Carlo estimation of the policy gradient. Also, our method does not depend on solving the high-dimensional Hamiltonian-Jacobi-Bellman equation or on any mesh discretization in the state and action space. These enable us to get rid of the curse of dimensionality and use this method in high-dimensional problems. Theoretically, we show that, under some assumptions, the M-CBO method can converge to the optimal control. In

*Table 3.* The value function $u(t, \mu)$ with $\mu$ being a mixture of three Gaussian random variables in the mean filed control problem.

| TIME | $n = 50$ | $n = 100$ | $n = 200$ | $n = 400$ | EXACT |
|------|----------|-----------|-----------|-----------|-------|
| 0.0 | 0.633 | 0.640 | 0.645 | 0.646 | 0.642 |
| 0.1 | 0.579 | 0.586 | 0.590 | 0.591 | 0.588 |
| 0.2 | 0.524 | 0.531 | 0.535 | 0.536 | 0.534 |
| 0.3 | 0.469 | 0.475 | 0.478 | 0.47 | 0.478 |
| 0.4 | 0.412 | 0.417 | 0.421 | 0.421 | 0.420 |
| 0.5 | 0.353 | 0.357 | 0.360 | 0.361 | 0.360 |
| 0.6 | 0.290 | 0.294 | 0.297 | 0.297 | 0.297 |
| 0.7 | 0.223 | 0.226 | 0.228 | 0.228 | 0.228 |
| 0.8 | 0.148 | 0.151 | 0.152 | 0.152 | 0.152 |
| 0.9 | 0.063 | 0.064 | 0.064 | 0.065 | 0.065 |

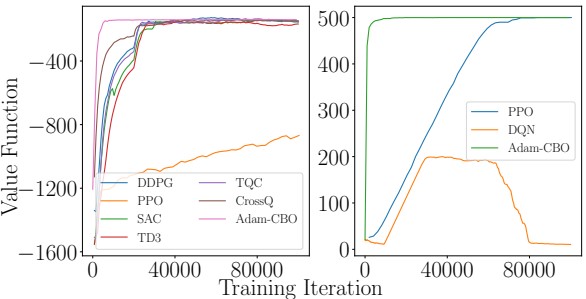

*Figure 9.* Comparison of value function convergence across methods. (Left) Performance on the Pendulum-v1 environment showing the evolution of the value function during training. (Right) Corresponding results for the CartPole-v1 environment. Our method demonstrates superior convergence in both cases.

the future, we are interested in applying our method to mean-field game problems and control problems with partial information and constraints (Ganapathi Subramanian et al., 2024; Hong & Tewari, 2024; Qiao & Wang, 2024; Sun et al., 2024; Wang et al., 2024).

## Acknowledgements

This work is partially supported by NSFC grant 12425113 and the Key Laboratory of the Ministry of Education for Mathematical Foundations and Applications of Digital Technology, University of Science and Technology of China.

## Impact Statement

This paper presents work whose goal is to advance the field of Machine Learning. There are many potential societal consequences of our work, none of which we feel must be specifically highlighted here.

*Table 4.* The computational time for each method over 100,000 steps

| METHOD | PENDULUM-V1 | CARTPOLE-V1 |
|--------|-------------|-------------|
| DDPG | 288.83 | |
| PPO | 145.19 | 150.58 |
| SAC | 355.01 | |
| TD3 | 291.26 | |
| TQC | 576.35 | |
| CROSSQ | 708.73 | |
| DQN | | 186.13 |
| ADAM-CBO | 1124.88 | 3444 |

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

# A. Well-posedness of the M-CBO method

In this section, we prove that the dynamics of the M-CBO method are well-posed. For an arbitrary but fixed $N$, we begin by studying the existence of a unique process $\left(\boldsymbol{\Theta}_t^{(N)}, \boldsymbol{\Omega}_t^{(N)}\right) = \left(\Theta^{(1,N)}, \cdots, \Theta^{(N,N)}, \Omega^{(1,N)}, \cdots, \Omega^{(N,N)}\right)$ that satisfies the M-CBO scheme (1)

$$
\begin{aligned}
\mathrm{d}\boldsymbol{\Theta}_t^{(N)} &= \mathbf{F}_{N,\Theta}\left(\boldsymbol{\Theta}_t^{(N)}, \boldsymbol{\Omega}_t^{(N)}\right)\mathrm{d}t + \sigma(t)\mathrm{d}\mathbf{W}_{\theta,t}^{(N)}, \\
\mathrm{d}\boldsymbol{\Omega}_t^{(N)} &= \mathbf{F}_{N,\Omega}\left(\boldsymbol{\Theta}_t^{(N)}, \boldsymbol{\Omega}_t^{(N)}\right)\mathrm{d}t + \sqrt{m}\sigma(t)\mathrm{d}\mathbf{W}_{\omega,t}^{(N)},
\end{aligned}
\tag{8}
$$

where $\mathbf{W}_{\theta,t}^{(N)}, \mathbf{W}_{\omega,t}^{(N)}$ is the standard Wiener process in $\mathbb{R}^{ND}$, and

$$
\begin{aligned}
\mathbf{F}_{N,\Theta}(\boldsymbol{\Theta},\boldsymbol{\Omega}) &= \left(F_{N,\Theta}^1(\boldsymbol{\Theta},\boldsymbol{\Omega}), \cdots F_{N,\Theta}^N(\boldsymbol{\Theta},\boldsymbol{\Omega})\right) \in \mathbb{R}^{ND}, \\
\mathbf{F}_{N,\Omega}(\boldsymbol{\Theta},\boldsymbol{\Omega}) &= \left(F_{N,\Omega}^1(\boldsymbol{\Theta},\boldsymbol{\Omega}), \cdots F_{N,\Omega}^N(\boldsymbol{\Theta},\boldsymbol{\Omega})\right) \in \mathbb{R}^{ND}, \\
F_{N,\Theta}^i(\boldsymbol{\Theta},\boldsymbol{\Omega}) &= \Omega^i - \gamma_1 \frac{\sum_{j\neq i}(\Theta^i - \Theta^j)w_\beta(\Theta^j)}{\sum_j w_\beta(\Theta^j)}, \\
F_{N,\Omega}^i(\boldsymbol{\Theta},\boldsymbol{\Omega}) &= -m\frac{\sum_{j\neq i}(\Theta^i - \Theta^j)w_\beta(\Theta^j)}{\sum_j w_\beta(\Theta^j)} + \gamma_2\Omega^i.
\end{aligned}
$$

Under the Assumption 4.1, we can easily deduce that $F_{N,\Theta}^i$ and $F_{N,\Omega}^i$ are locally Lipschitz continuous and have linear growth. Consequently, $(\mathbf{F}_{N,\Theta}, \mathbf{F}_{N,\Omega})$ is locally Lipschitz continuous and has linear growth. More precisely, we have the following lemma.

**Lemma A.1.** *Let $N \in \mathbb{N}$, $\beta, R > 0$ be arbitrary. Then for any $(\boldsymbol{\Theta}, \boldsymbol{\Omega}), (\hat{\boldsymbol{\Theta}}, \hat{\boldsymbol{\Omega}}) \in \mathbb{R}^D \times \mathbb{R}^D$ with $\|\boldsymbol{\Theta}\| + \|\boldsymbol{\Omega}\|, \|\hat{\boldsymbol{\Theta}}\| + \|\hat{\boldsymbol{\Omega}}\| \leq R$ and all $i = 1, \cdots, N$, it holds*

$$
\|F_{N,\Theta}^i(\boldsymbol{\Theta},\boldsymbol{\Omega}) - F_{N,\Theta}^i(\hat{\boldsymbol{\Theta}},\hat{\boldsymbol{\Omega}})\| \leq \gamma_1\|\Theta^i - \hat{\Theta}^i\| + \|\Omega^i - \hat{\Omega}^i\| + \gamma_1\left(1 + 2\frac{c_R}{N}\sqrt{N\|\hat{\Theta}^i\|^2 + \|\hat{\boldsymbol{\Theta}}\|^2}\right)\|\boldsymbol{\Theta} - \hat{\boldsymbol{\Theta}}\|,
$$

$$
\|F_{N,\Omega}^i(\boldsymbol{\Theta},\boldsymbol{\Omega}) - F_{N,\Omega}^i(\hat{\boldsymbol{\Theta}},\hat{\boldsymbol{\Omega}})\| \leq \|\Theta^i - \hat{\Theta}^i\| + \gamma_2\|\Omega^i - \hat{\Omega}^i\| + m\left(1 + 2\frac{c_R}{N}\sqrt{N\|\hat{\Theta}^i\|^2 + \|\hat{\boldsymbol{\Theta}}\|^2}\right)\|\boldsymbol{\Theta} - \hat{\boldsymbol{\Theta}}\|,
$$

$$
\|F_{N,\Theta}^i(\boldsymbol{\Theta},\boldsymbol{\Omega})\| \leq \gamma_1\|\Theta^i\| + \|\Omega^i\| + \gamma_1\|\boldsymbol{\Theta}\|,
$$

$$
\|F_{N,\Omega}^i(\boldsymbol{\Theta},\boldsymbol{\Omega})\| \leq m\|\Theta^i\| + \gamma_2\|\Omega^i\| + m\|\boldsymbol{\Theta}\|,
$$

*where $c_R = \alpha\|\nabla\mathcal{J}\|_{L^\infty(B_{\theta,R}(0))}\exp\left(\beta\|\mathcal{J} - \underline{\mathcal{J}}\|_{L^\infty(B_{\theta,R}(0))}\right)$.*

*Proof.* From Lemma 2.1 (Carrillo et al., 2018), we have

$$
\left\|\frac{\sum_{j\neq i}(\Theta^i - \Theta^j)w_\beta(\Theta^j)}{\sum_j w_\beta(\Theta^j)} - \frac{\sum_{j\neq i}(\hat{\Theta}^i - \hat{\Theta}^j)w_\beta(\hat{\Theta}^j)}{\sum_j w_\beta(\hat{\Theta}^j)}\right\| \leq \|\Theta^i - \hat{\Theta}^i\| + \left(1 + 2\frac{c_R}{N}\sqrt{N\|\hat{\Theta}^i\|^2 + \|\hat{\boldsymbol{\Theta}}\|^2}\right)\|\boldsymbol{\Theta} - \hat{\boldsymbol{\Theta}}\|,
$$

$$
\left\|\frac{\sum_{j\neq i}(\Theta^i - \Theta^j)w_\beta(\Theta^j)}{\sum_j w_\beta(\Theta^j)}\right\| \leq \|\Theta^i\| + \|\boldsymbol{\Theta}\|.
$$

By the triangle inequality, the required estimation is proved. $\square$

Based on Lemma A.1, we may invoke standard existence results of strong solutions for Equation (1).

*Proof of Theorem 4.2.* We make use of the standard result on the existence of a unique strong solution here. To this end, we show the existence $b_N > 0$, such that

$$
\boldsymbol{\Theta} \cdot \mathbf{F}_{N,\Theta}(\boldsymbol{\Theta},\boldsymbol{\Omega}) + \boldsymbol{\Omega} \cdot \mathbf{F}_{N,\Omega}(\boldsymbol{\Theta},\boldsymbol{\Omega}) + N(m+1)D\sigma(t)^2 \leq b_N(\|\boldsymbol{\Theta}\|^2 + \|\boldsymbol{\Omega}\|^2 + 1).
$$

Notice that

$$-\Theta^i\frac{\sum_{j\neq i}(\Theta^i-\Theta^j)w_\beta(\Theta^j)}{\sum_j w_\beta(\Theta^j)} \leq -\|\Theta^i\|^2 + \|\Theta^i\|\|\mathbf{\Theta}\|,$$

$$-\Omega^i\frac{\sum_{j\neq i}(\Theta^i-\Theta^j)w_\beta(\Theta^j)}{\sum_j w_\beta(\Theta^j)} \leq \|\Omega^i\|\|\Theta^i\| + \|\Omega^i\|\|\mathbf{\Theta}\|,$$

we have the following inequalities

$$
\begin{aligned}
\Theta^i F_{N,\Theta}^i(\mathbf{\Theta},\mathbf{\Omega}) &= \Theta^i\Omega^i - \gamma_1\Theta^i\frac{\sum_{j\neq i}(\Theta^i-\Theta^j)w_\beta(\Theta^j)}{\sum_j w_\beta(\Theta^j)} \\
&\leq \frac{1}{2}\|\Theta^i\|^2 + \frac{1}{2}\|\Omega^i\|^2 - \gamma_1\|\Theta^i\|^2 + \gamma_1\|\Theta^i\|\|\mathbf{\Theta}\| \\
&\leq \frac{1}{2}\|\Theta^i\|^2 + \frac{1}{2}\|\Omega^i\|^2 + \gamma_1\|\Theta^i\|\|\mathbf{\Theta}\| \\
&\leq \left(\frac{1}{2}+\frac{\gamma_1}{2}\right)\|\Theta^i\|^2 + \frac{\gamma_1}{2}\|\mathbf{\Theta}\|^2 + \frac{1}{2}\|\Omega^i\|^2,
\end{aligned}
$$

and

$$
\begin{aligned}
\Omega^i F_{N,\Omega}^i(\mathbf{\Theta},\mathbf{\Omega}) &= -m\Omega^i\frac{\sum_{j\neq i}(\Theta^i-\Theta^j)w_\beta(\Theta^j)}{\sum_j w_\beta(\Theta^j)} + \gamma_2\|\Omega^i\|^2 \\
&\leq m\|\Omega^i\|\|\Theta^i\| + m\|\Omega^i\|\|\mathbf{\Theta}\| + \gamma_2\|\Omega^i\|^2 \\
&\leq (m+\gamma_2)\|\Omega^i\|^2 + \frac{m}{2}\|\Theta^i\|^2 + \frac{m}{2}\|\mathbf{\Theta}\|^2.
\end{aligned}
$$

Therefore, we conclude that

$$
\begin{aligned}
&\mathbf{\Theta}\cdot\mathbf{F}_{N,\Theta}(\mathbf{\Theta},\mathbf{\Omega}) + \mathbf{\Omega}\cdot\mathbf{F}_{N,\Omega}(\mathbf{\Theta},\mathbf{\Omega}) + (m+1)D\sigma(t)^2 \\
&\leq N(m+1)D\sigma(t)^2 + \sum_{i=1}^N\left(\Theta^i\cdot F_{N,\Theta}^i(\mathbf{\Theta},\mathbf{\Omega}) + \Omega^i\cdot F_{N,\Omega}^i(\mathbf{\Theta},\mathbf{\Omega})\right) \\
&\leq N(m+1)D\sigma(t)^2 + \sum_{i=1}^N\left(\left(\frac{1}{2}+\frac{\gamma_1}{2}\right)\|\Theta^i\|^2 + \frac{\gamma_1}{2}\|\mathbf{\Theta}\|^2 + \frac{1}{2}\|\Omega^i\|^2 + (m+\gamma_2)\|\Omega^i\|^2 + \frac{m}{2}\|\Theta^i\|^2 + \frac{m}{2}\|\mathbf{\Theta}\|^2\right) \\
&\leq N(m+1)D\sigma(t)^2 + \left(\frac{1+\gamma_1+m}{2}+\frac{\gamma_1+N}{2}\right)\|\mathbf{\Theta}\|^2 + \left(m+\gamma_2+\frac{1}{2}\right)\|\mathbf{\Omega}\|^2 \\
&\leq b_N(\|\mathbf{\Theta}\|^2 + \|\mathbf{\Omega}\|^2 + 1),
\end{aligned}
$$

$$(9)$$

where $b_N = \max\{N(m+1)D\bar{\sigma}^2, \frac{1+\gamma_1+m}{2}+\frac{\gamma_1+N}{2}, m+\gamma_2+\frac{1}{2}\} > 0$. Then we apply Theorem 3.1 in (Durrett, 2018) to finish the existence and uniqueness proof. $\qquad\square$

## B. Well-posedness of the Mean Field Equations

**Definition B.1.** We say $\rho_t \in \mathcal{C}\left([0,T], \mathcal{P}_4(\mathbb{R}^D\times\mathbb{R}^D)\right)$ is a weak solution to the Fokker-Planck equation (3) with initial condition $\rho_0$, if $\forall\phi\in\mathcal{C}\left(\mathbb{R}^D\times\mathbb{R}^D\right)$, we have

$$
\begin{aligned}
\frac{\mathrm{d}}{\mathrm{d}t}\int\phi(\theta,\omega)\mathrm{d}\rho_t &= \int\langle\omega-\gamma_1(\theta-\mathcal{M}_\beta[\mu_t]),\nabla_\theta\phi\rangle\mathrm{d}\rho_t \\
&\quad -\int\langle m(\theta-\mathcal{M}_\beta[\mu_t])+\gamma_2\omega,\nabla_\omega\phi\rangle\mathrm{d}\rho_t \\
&\quad +\frac{m\sigma(t)^2}{2}\int\Delta_\omega\phi\mathrm{d}\rho_t + \frac{\sigma(t)^2}{2}\int\Delta_\theta\phi\mathrm{d}\rho_t,
\end{aligned}
$$

$$(10)$$

and $\lim_{t\to\infty}\rho_t = \rho_0$ in a pointwise sense.

To prove Theorem 4.3, we start with the following lemma.

**Lemma B.2.** *If $\mathcal{J}$ satisfies Assumption 4.1 and $\rho, \hat{\rho} \in \mathcal{P}_2(\mathbb{R}^D \times \mathbb{R}^D)$ with*

$$\int \|\theta\|^4 + \|\omega\|^4 \mathrm{d}\rho, \int \|\hat{\theta}\|^4 + \|\hat{\omega}\|^4 \mathrm{d}\hat{\rho} \leq K,$$

*then the following stability estimate holds*

$$|\mathcal{M}_\beta[\mu] - \mathcal{M}_\beta[\hat{\mu}]| \leq c_0 W_2(\rho, \hat{\rho})$$

*for a constant $c_0 > 0$ depending on $\beta, L_J$ and $K$, where $\mu(\theta) = \int_{\mathbb{R}^D} \rho(\theta, \mathrm{d}\omega), \hat{\mu}(\hat{\theta}) = \int_{\mathbb{R}^D} \hat{\rho}(\hat{\theta}, \mathrm{d}\hat{\omega})$.*

*Proof.* By Lemma 3.2 in (Carrillo et al., 2018), we have $|\mathcal{M}_\beta[\mu] - \mathcal{M}_\beta[\hat{\mu}]| \leq c_0 W_2(\mu, \hat{\mu}) = c_0 \inf \mathbb{E}_{(\mu, \hat{\mu})}[\|\theta - \hat{\theta}\|^2]$. Therefore, we have $\inf E_{(\mu, \hat{\mu})}[\|\theta - \hat{\theta}\|^2] \leq \inf \mathbb{E}_{(\rho, \hat{\rho})}[\|\theta - \hat{\theta}\|^2] + \inf \mathbb{E}_{(\rho, \hat{\rho})}[\|\omega - \hat{\omega}\|^2] \leq \inf \mathbb{E}_{(\rho, \hat{\rho})}[\|\theta - \hat{\theta}\|^2 + \|\omega - \hat{\omega}\|^2] = W_2(\rho, \hat{\rho})$. We prove the boundedness here. □

To prove the existence and uniqueness, we recall the Leray-Schauder fixed point theorem (Theorem 11.3 in (Gilbarg et al., 2001)).

**Theorem B.3.** *Let $T$ be a compact mapping of a Banach space $\mathcal{B}$ into itself, and suppose there exists a constant $M$ such that $\|x\|_{\mathcal{B}} \leq M$ for all $x \in \mathcal{B}$ and $\eta \in (0, 1)$ satisfying $x = \eta Tx$. Then $T$ has a fixed point.*

*Proof of Theorem 4.3.* **Step 1 (Construction of map $T$)**

Let us fix $u_t \in \mathcal{C}([0, T])$. By Theorem 6.2.2 in (Arnold, 1976), there is a unique solution to

$$\mathrm{d}\theta_t = \omega_t \mathrm{d}t - \gamma_1(\theta_t - u_t)\mathrm{d}t + \sigma(t)\mathrm{d}W_{\theta,t},$$
$$\mathrm{d}\omega_t = -m(\theta_t - u_t)\mathrm{d}t - \gamma_2\omega_t\mathrm{d}t + \sqrt{m}\sigma(t)\mathrm{d}W_{\omega,t},$$

where $(\theta_0, \omega_0) \sim \rho_0$. We use $\rho_t$ to denote the corresponding law of the unique solution. Using $\rho_t$, one can compute $\mu_t(\theta) = \int \rho_t(\theta, \mathrm{d}\omega)$ and $\mathcal{M}_\beta[\mu_t]$, which is uniquely determined by $u_t$ and is in $\mathcal{C}([0, T])$. Thus, one can construct a map from $\mathcal{C}([0, T])$ to $\mathcal{C}([0, T])$, which maps $u_t$ to $\mathcal{M}_\beta[\mu_t]$.

**Step 2 (Compactness)** First, by referencing Chapter 7 in (Arnold, 1976), we obtain the inequality for the solution $\theta_t, \omega_t$ to Equation (B):

$$\mathbb{E}[\|\theta_t\| + \|\omega_t\|]^4 \leq \left(1 + \mathbb{E}[\|\theta_0\| + \|\omega_0\|]^4\right)\exp(ct),$$

where $c > 0$. Thus one can deduce $\mathbb{E}[\|\theta_t\|^4 + \|\omega_t\|^4] \lesssim 1$ and $\mathbb{E}[\|\theta_t\|^2 + \|\omega_t\|^2] \lesssim 1$.

By Lemma B.2, we have $\|\mathcal{M}_\beta(\mu_t) - \mathcal{M}_\beta(\mu_s)\| \leq c_0 W_2(\rho_t, \rho_s)$. For $W_2(\rho_t, \rho_s)$, it holds that $W_2(\rho_t, \rho_s) \leq E[\|\theta_t - \theta_s\| + \|\omega_t - \omega_s\|] \leq \sqrt{2E[\|\theta_t - \theta_s\|^2 + 2\|\omega_t - \omega_s\|^2]}$. Further, we can deduce

$$\theta_t - \theta_s = \int_s^t \omega_\tau - \gamma_1(\theta_\tau - u_\tau)\mathrm{d}\tau + \int_s^t \sigma(\tau)\mathrm{d}W_{\theta,\tau},$$
$$\omega_t - \omega_s = \int_s^t -m(\theta_\tau - u_\tau) - \gamma_2\omega_\tau\mathrm{d}\tau + \sqrt{m}\int_s^t \sigma(\tau)\mathrm{d}W_{\omega,\tau}.$$

Thus

$$\mathbb{E}[\|\theta_t - \theta_s\|^2 + \|\omega_t - \omega_s\|^2] \lesssim \mathbb{E}\left[\left\|\int_s^t \omega_\tau\mathrm{d}\tau\right\|^2\right] + \mathbb{E}\left[\left\|\int_s^t (\theta_\tau - u_\tau)\mathrm{d}\tau\right\|^2\right]$$
$$+ \mathbb{E}\left[\left\|\int_s^t \sigma(\tau)\mathrm{d}W_{\theta,\tau}\right\|^2\right] + m\mathbb{E}\left[\left\|\int_s^t \sigma(\tau)\mathrm{d}W_{\omega,\tau}\right\|^2\right].$$

Let us proceed to bound the four terms on the right-hand side individually. Consider the first term, where we establish

$$\mathbb{E}\left[\left\|\int_s^t \omega_\tau\mathrm{d}\tau\right\|^2\right] \leq \mathbb{E}\left[\left(\int_s^t \|\omega_\tau\|\mathrm{d}\tau\right)^2\right]$$
$$\leq |t - s|\mathbb{E}\left[\int_s^t \|\omega_\tau\|^2\mathrm{d}\tau\right] \lesssim |t - s|.$$

The first inequality in this sequence is derived from the Cauchy-Schwarz inequality, followed by an application of Jensen's inequality for the second inequality. The final inequality is attributed to the boundedness property of the solution. Similarly, for the second term, we have

$$\mathbb{E}\left[\left\|\int_s^t (\theta_\tau - u_\tau)\mathrm{d}\tau\right\|^2\right] \le \mathbb{E}\left[\left(\int_s^t \|\theta_\tau - u_\tau\|\,\mathrm{d}\tau\right)^2\right]$$

$$\le |t - s|\mathbb{E}\left[\int_s^t \|\theta_\tau - u_\tau\|^2\,\mathrm{d}\tau\right]$$

$$\lesssim |t - s|\left(\mathbb{E}\left[\int_s^t \|\theta_\tau\|^2\,\mathrm{d}\tau + \int_s^t \|u_\tau\|^2\,\mathrm{d}\tau\right]\right) \lesssim |t - s|.$$

For the third and fourth terms, we use the Itô Isometry,

$$\mathbb{E}\left[\left\|\int_s^t \sigma(\tau)\mathrm{d}W_{\omega,\tau}\right\|^2\right] = \mathbb{E}\left[\left\|\int_s^t \sigma(\tau)\mathrm{d}W_{\theta,\tau}\right\|^2\right] = \mathbb{E}\left[\int_s^t \sigma(\tau)^2\mathrm{d}\tau\right] \le \bar{\sigma}^2|t - s|.$$

Finially, we combine the inequality to deduce that $\|\mathcal{M}_\beta[\mu_t] - \mathcal{M}_\beta[\mu_s]\| \lesssim |t - s|^{1/2}$, which implies that $\mathcal{M}_\beta(\mu_t) \in \mathcal{C}^{0,1/2}[0, T]$. Thus $T$ is compact.

**Step 3 (Existence)** We make use of Theorem B.3. Take $u_t = \eta T u_t$ for $\eta \in [0, 1]$. We now try to prove $\|u_t\|_\infty \le q$ for some finite $q > 0$. First, one has

$$\|u_t\|^2 = \eta^2 \|\mathcal{M}_\beta(\mu_t)\|^2 \le \eta^2 \exp(\beta(\overline{J} - \underline{J}))\int \|\theta\|^2\mathrm{d}\rho_t \le \eta^2 \exp(\beta(\overline{J} - \underline{J}))\int \|\theta\|^2 + m^{-1}\|\omega\|^2\mathrm{d}\rho_t.$$

Then we try to prove the boundedness of $\int \|\theta\|^2 + m^{-1}\|\omega\|^2\mathrm{d}\rho_t$. Since $\rho_t$ is a weak solution of the Fokker-Planck equation, one has

$$\frac{\mathrm{d}}{\mathrm{d}t}\int \left(\|\theta\|^2 + m^{-1}\|\omega\|^2\right)\mathrm{d}\rho_t = \int \omega \cdot \theta - \gamma_1(\theta - u_t)\cdot\theta - (\theta - u_t)\cdot\omega - \gamma m^{-1}\omega\cdot\omega\mathrm{d}\rho_t$$

$$= \int -\gamma_1(\theta - u_t)\cdot\theta + u_t\cdot\omega - \gamma m^{-1}\omega\cdot\omega\mathrm{d}\rho_t$$

Since

$$\int \theta\cdot u_t\mathrm{d}\rho_t \lesssim \int \|\theta\|^2 + \|u_t\|^2\mathrm{d}\rho_t \lesssim \int \|\theta\|^2\mathrm{d}\rho_t + \int (\|\theta\|^2 + m^{-1}\|\omega\|^2)\mathrm{d}\rho_t,$$

and

$$\int \omega\cdot u_t\mathrm{d}\rho_t \lesssim \int \|\omega\|^2 + \|u_t\|^2\mathrm{d}\rho_t \lesssim \int \|\omega\|^2\mathrm{d}\rho_t + \int (\|\theta\|^2 + m^{-1}\|\omega\|^2)\mathrm{d}\rho_t,$$

we can deduce that

$$\frac{\mathrm{d}}{\mathrm{d}t}\int \left(\|\theta\|^2 + m^{-1}\|\omega\|^2\right)\mathrm{d}\rho_t \lesssim \left(\|\theta\|^2 + m^{-1}\|\omega\|^2\right)\mathrm{d}\rho_t.$$

Applying Gronwall's inequality yields that $\int \left(\|\theta\|^2 + m^{-1}\|\omega\|^2\right)\mathrm{d}\rho_t$ is bounded and the above inequality is independent of $u_t$. Thus we have shown that $\|u_t\|_\infty$ is bounded by a uniform constant $q$. Theorem B.3 then gives the existence.

**Step 4(Uniqueness):** Suppose we are given two fixed points of $T$: $u_t$ and $\hat{u}_t$ with $\|u\|_\infty, \|\hat{u}\|_\infty \le q$ and $\sup_{t\in[0,T]}\int \|\theta\|^4 + \|\omega\|^4\mathrm{d}\rho_t, \sup_{t\in[0,T]}\int \|\hat{\theta}\|^4 + \|\hat{\omega}\|^4\mathrm{d}\hat{\rho}_t \le K$ and their corresponding process $(\Theta, \Omega), (\hat{\Theta}, \hat{\Omega})$ satisfying respectively. Then take the difference $\delta\Theta := \Theta - \hat{\Theta}$ and $\delta\Omega := \Omega - \hat{\Omega}$. One has

$$\delta\Theta_t = \delta\Theta_0 + \int_0^t \delta\Omega_\tau\mathrm{d}\tau - \gamma_1\int_0^t \delta\Theta_\tau\mathrm{d}\tau + \gamma_1\int_0^t (u_\tau - \hat{u}_\tau)\mathrm{d}\tau,$$

$$\delta\Omega_t = \delta\Omega_0 - \gamma_2\int_0^t \delta\Omega_\tau\mathrm{d}\tau - m\int_0^t \delta\Theta_\tau\mathrm{d}\tau + m\int_0^t (u_\tau - \hat{u}_\tau)\mathrm{d}\tau.$$

Thus

$$\mathbb{E}[\|\delta\Theta_t\|^2 + \|\delta\Omega_t\|^2] \lesssim \mathbb{E}[\|\delta\Theta_0\|^2 + \|\delta\Omega_0\|^2] + \mathbb{E}\left[\left(\int_0^t \|\delta\Omega_\tau\|\,\mathrm{d}\tau\right)^2\right] + \mathbb{E}\left[\left(\int_0^t \|\delta\Theta_\tau\|\,\mathrm{d}\tau\right)^2\right]$$
$$+ \mathbb{E}\left[\left(\int_0^t \|u_\tau - \hat{u}_\tau\|\,\mathrm{d}\tau\right)^2\right].$$

For the $\mathbb{E}\left[\left(\int_0^t \|u_\tau - \hat{u}_\tau\|\,\mathrm{d}\tau\right)^2\right]$, we have that

$$\mathbb{E}\left[\left(\int_0^t \|u_\tau - \hat{u}_\tau\|\,\mathrm{d}\tau\right)^2\right] = \mathbb{E}\left[\left(\int_0^t \|\mathcal{M}_\beta[\mu_\tau] - \mathcal{M}_\beta[\hat{\mu}_\tau]\|\,\mathrm{d}\tau\right)^2\right]$$
$$\leq t\mathbb{E}\left[\int_0^t \|\mathcal{M}_\beta[\mu_\tau] - \mathcal{M}_\beta[\hat{\mu}_\tau]\|^2\,\mathrm{d}\tau\right].$$

Thus we have

$$\mathbb{E}[\|\delta\Theta_t\|^2 + \|\Omega_t\|^2] \lesssim \mathbb{E}[\|\delta\Theta_0\|^2 + \|\delta\Omega_0\|^2] + \mathbb{E}\left[\left(\int_0^t \|\delta\Omega_\tau\|\,\mathrm{d}\tau\right)^2\right] + \mathbb{E}\left[\left(\int_0^t \|\delta\Theta_\tau\|\,\mathrm{d}\tau\right)^2\right]$$
$$+ \mathbb{E}\left[\int_0^t \|\mathcal{M}_\beta[\mu_\tau] - \mathcal{M}_\beta[\hat{\mu}_\tau]\|^2\,\mathrm{d}\tau\right].$$

Notice that by Lemma B.2, $\|\mathcal{M}_\beta[\mu_\tau] - \mathcal{M}_\beta[\hat{\mu}_\tau]\| \lesssim W_2(\rho_\tau, \hat{\rho}_\tau) \leq \sqrt{\mathbb{E}\left[\|\delta\Theta_\tau\|^2 + \|\delta\Omega_\tau\|^2\right]}$. So we can deduce

$$\mathbb{E}\left[\|\delta\Theta_\tau\|^2 + \|\delta\Omega_\tau\|^2\right] \lesssim \mathbb{E}\left[\|\delta\Theta_0\|^2 + \|\delta\Omega_0\|^2\right] + \mathbb{E}\left[\int_0^t \|\delta\Omega_\tau\|^2 + \|\delta\Theta_\tau\|^2\,\mathrm{d}\tau\right].$$

By the Gronwall'sinequality with the fact that $\mathbb{E}[\|\delta\Theta_0\|^2 + \|\delta\Omega_0\|^2] = 0$ gives that uniqueness result. $\qquad\square$

## C. Mean Field Limit

In this section, we prove the connection between the solution to Equation (1) and the solution of the Fokker-Planck equation (3). We begin with the following boundedness result.

**Lemma C.1.** *Let $\mathcal{J}$ satisfy Assumption 4.1 and $\rho_0 \in \mathcal{P}_4(\mathbb{R}^D \times \mathbb{R}^D)$. For any $N \geq 2$, assume that $\{(\Theta_t^{(i,N)}, \Omega_t^{(i,N)})_{t\in[0,T]}\}_{i=1}^N$ is the unique solution to Equation (8) with $\rho_0^{\otimes N}$-distributed initial data $\{(\Theta_0^{(i,N)}, \Omega_0^{(i,N)})\}_{i=1}^N$. Then there exists a constant $K > 0$ independent of $N$ such that*

$$\sup_{i=1\cdots N}\left\{\sup_{t\in[0,T]} \mathbb{E}\left[\|\Theta_t^{(i,N)}\|^2 + \|\Omega_t^{(i,N)}\|^2\right]\right\} \leq K,$$

$$\sup_{i=1\cdots N}\left\{\sup_{t\in[0,T]} \mathbb{E}\left[\|\Theta_t^{(i,N)}\|^4 + \|\Omega_t^{(i,N)}\|^4\right]\right\} \leq K,$$

$$\sup_{i=1\cdots N}\left\{\sup_{t\in[0,T]} \mathbb{E}\left[\|\mathcal{M}_\beta[\hat{\mu}_t^N]\|^2\right]\right\} \leq K,$$

$$\sup_{i=1\cdots N}\left\{\sup_{t\in[0,T]} \mathbb{E}\left[\|\mathcal{M}_\beta[\hat{\mu}_t^N]\|^4\right]\right\} \leq K,$$

*where $\hat{\mu}_t^N = \frac{1}{N}\sum_{i=1}^N \delta_{\Theta_t^{(i,N)}}$ is the empirical measure.*

*Proof.* For each $i$, we have

$$\mathrm{d}\Theta_t^{(i,N)} = -\Omega_t^{(i,N)}\mathrm{d}t - \gamma_1(\Theta_t^{(i,N)} - \mathcal{M}_\beta[\hat{\mu}_t^N])\mathrm{d}t + \sigma(t)\mathrm{d}W_{\theta,t}^i,$$
$$\mathrm{d}\Omega_t^{(i,N)} = -m(\Theta_t^{(i,N)} - \mathcal{M}_\beta[\hat{\mu}_t^N])\mathrm{d}t - \gamma_2\Omega_t^{(i,N)}\mathrm{d}t + \sqrt{m}\sigma(t)\mathrm{d}W_{\omega,t}^i.$$

Now we pick $p = 1$ or $p = 2$. Then

$$
\begin{aligned}
\mathbb{E}\left[\left\|\Omega_t^{(i,N)}\right\|^{2p}\right] + \mathbb{E}\left[\left\|\Theta_t^{(i,N)}\right\|^{2p}\right] \lesssim & \mathbb{E}\left[\left\|\Theta_0^{(i,N)}\right\|^{2p}\right] + \mathbb{E}\left[\left\|\Omega_0^{(i,N)}\right\|^{2p}\right] \\
& + \mathbb{E}\left[\int_0^t \left\|\Theta_\tau^{(i,N)}\right\| \mathrm{d}\tau\right]^{2p} + \mathbb{E}\left[\int_0^t \left\|\Omega_\tau^{(i,N)}\right\| \mathrm{d}\tau\right]^{2p} \\
& + \mathbb{E}\left[\int_0^t \left\|\mathcal{M}_\beta[\hat{\mu}_\tau^N]\right\| \mathrm{d}\tau\right]^{2p} + \mathbb{E}\left[\int_0^t \sigma(\tau)\mathrm{d}W_{\theta,\tau}\right]^{2p}.
\end{aligned}
$$

Now apply the Cauchy's inequality,

$$
\mathbb{E}\left[\int_0^t \|\Theta_\tau^{(i,N)}\|\mathrm{d}\tau\right]^{2p} \leq t^p \cdot \mathbb{E}\left[\int_0^t \|\Theta_\tau^{(i,N)}\|^2\mathrm{d}\tau\right]^p, \quad \mathbb{E}\left[\int_0^t \|\Omega_\tau^{(i,N)}\|\mathrm{d}\tau\right]^{2p} \leq t^p \cdot \mathbb{E}\left[\int_0^t \|\Omega_\tau^{(i,N)}\|^2\mathrm{d}\tau\right]^p,
$$

and

$$
\mathbb{E}\left[\int_0^t \|\mathcal{M}_\beta[\hat{\mu}_\tau^N]\|\mathrm{d}\tau\right]^{2p} \leq t^p \mathbb{E}\left[\int_0^t \|\mathcal{M}_\beta[\hat{\mu}_\tau^N]\|^2\mathrm{d}\tau\right]^p.
$$

Also, by Itô Isometry,

$$
\mathbb{E}\left[\int_0^t \sigma(\tau)\mathrm{d}W_{\theta,\tau}\right]^{2p} = \mathbb{E}\left[\int_0^t \sigma(\tau)^2\mathrm{d}\tau\right]^p.
$$

Thus

$$
\begin{aligned}
\mathbb{E}\left[\left\|\Omega_t^{(i,N)}\right\|^{2p} + \left\|\Theta_t^{(i,N)}\right\|^{2p}\right] \lesssim & \mathbb{E}\left[\left\|\Omega_0^{(i,N)}\right\|^{2p}\right] + \mathbb{E}\left[\left\|\Theta_0^{(i,N)}\right\|^{2p}\right] \\
& + \mathbb{E}\left[\int_0^t \left\|\Theta_\tau^{(i,N)}\right\|^2 \mathrm{d}\tau\right]^p + \mathbb{E}\left[\int_0^t \left\|\Omega_\tau^{(i,N)}\right\|^2 \mathrm{d}\tau\right]^p \\
& + \mathbb{E}\left[\int_0^t \left\|\mathcal{M}_\beta[\hat{\mu}_\tau]\right\|^2 \mathrm{d}\tau\right]^p + 1.
\end{aligned}
$$

Further, by Hölder inequality,

$$
\mathbb{E}\left[\int_0^t \left\|\Theta_\tau^{(i,N)}\right\|^2 \mathrm{d}\tau\right]^p \leq \mathbb{E}\left[\int_0^t \left\|\Theta_\tau^{(i,N)}\right\|^{2p} \mathrm{d}\tau\right], \quad \mathbb{E}\left[\int_0^t \|\Omega_\tau^{(i,N)}\|^2\mathrm{d}\tau\right]^p \leq \mathbb{E}\left[\int_0^t \|\Omega_\tau^{(i,N)}\|^{2p}\mathrm{d}\tau\right]
$$

and

$$
\mathbb{E}\left[\int_0^t \left\|M_\beta[\hat{\mu}_t^N]\right\|^2 \mathrm{d}\tau\right]^p \leq \mathbb{E}\left[\int_0^t \left\|M_\beta[\hat{\mu}_t^N]\right\|^{2p} \mathrm{d}\tau\right].
$$

So we can deduce

$$
\begin{aligned}
\mathbb{E}\left[\left\|\Omega_t^{(i,N)}\right\|^{2p} + \left\|\Theta_t^{(i,N)}\right\|^{2p}\right] \lesssim & \mathbb{E}\left[\left\|\Omega_0^{(i,N)}\right\|^{2p}\right] + \mathbb{E}\left[\left\|\Theta_0^{(i,N)}\right\|^{2p}\right] \\
& + \mathbb{E}\left[\int_0^t \left\|\Theta_\tau^{(i,N)}\right\|^{2p} \mathrm{d}\tau\right] + \mathbb{E}\left[\int_0^t \left\|\Omega_\tau^{(i,N)}\right\|^{2p} \mathrm{d}\tau\right] \\
& + \mathbb{E}\left[\int_0^t \|\mathcal{M}_\beta[\hat{\mu}_\tau^N]\|^{2p}\mathrm{d}\tau\right] + 1.
\end{aligned}
$$

Thus

$$
\begin{aligned}
\mathbb{E}\left[\int \left(\|\theta\|^{2p} + \|\omega\|^{2p}\right)\mathrm{d}\hat{\rho}_t^N\right] \lesssim & \mathbb{E}\left[\int \left(\|\theta\|^{2p} + \|\omega\|^{2p}\right)\mathrm{d}\hat{\rho}_0^N\right] \\
& + \int_0^t \left(\mathbb{E}\left[\int \left(\|\theta\|^{2p} + \|\omega\|^{2p}\right)\mathrm{d}\hat{\rho}_\tau^N\right]\right)\mathrm{d}\tau \\
& + \int_0^t \mathbb{E}\left[\|\mathcal{M}_\beta[\hat{\mu}_\tau^N]\|\right]^{2p}\mathrm{d}\tau + 1.
\end{aligned}
$$

It follows from Lemma 3.1 in (Carrillo et al., 2018), we have

$$\int \|\theta\|^2 \frac{w_\beta(\theta)}{\|w_\beta\|_{L^1(\hat\rho_\tau^N)}} \mathrm{d}\hat\rho_\tau^N \le b_1 + b_2 \int \|\theta\|^2 \mathrm{d}\hat\rho_\tau^N \le b_1 + b_2 \int (\|\theta\|^2 + \|\omega\|^2)\mathrm{d}\hat\rho_\tau^N.$$

Then we can calculate

$$
\begin{aligned}
\|M_\beta[\hat\mu_t^N]\|^{2p} &= \left\| \int \theta \cdot \frac{w_\beta(\theta)}{\|w_\beta\|_{L^1(\hat\rho_\tau^N)}} \mathrm{d}\hat\rho_\tau^N \right\|^{2p} \\
&\le \left( \int \|\theta\| \frac{w_\beta(\theta)}{\|w_\beta\|_{L^1(\hat\rho_\tau^N)}} \mathrm{d}\hat\rho_\tau^N \right)^{2p} \\
&\le \left( \int \|\theta\|^2 \frac{w_\beta(\theta)}{\|w_\beta\|_{L^1(\hat\rho_\tau^N)}} \frac{w_\beta(\theta)}{\|w_\beta\|_{L^1(\hat\rho_\tau^N)}} \mathrm{d}\hat\rho_\tau^N \right)^p \\
&\lesssim \left( \int \|\theta\|^2 \frac{w_\beta(\theta)}{\|w_\beta\|_{L^1(\hat\rho_\tau^N)}} \mathrm{d}\hat\rho_\tau^N \right)^p \\
&\le \left( b_1 + b_2 \int (\|\theta\|^2 + \|\omega\|^2)\mathrm{d}\hat\rho_\tau^N \right)^{2p} \\
&\lesssim 1 + \int (\|\theta\|^{2p} + \|\omega\|^{2p})\mathrm{d}\hat\rho_\tau^N.
\end{aligned}
$$

Combining the above inequality leads to

$$
\begin{aligned}
\mathbb{E}\left[ \int (\|\theta\|^{2p} + \|\omega\|^{2p})\mathrm{d}\hat\rho_t^N \right] \lesssim & \mathbb{E}\left[ \int (\|\theta\|^{2p} + \|\omega\|^{2p})\,\mathrm{d}\hat\rho_0^N \right] \\
& + \int_0^t \left( \mathbb{E}\left[ \int (\|\theta\|^{2p} + \|\omega\|^{2p})\,\mathrm{d}\hat\rho_\tau^N \right] \right) \mathrm{d}\tau + 1.
\end{aligned}
$$

By applying Gronwall's inequality, it follows that $\mathbb{E}\left[ \int (\|\theta\|^{2p} + \|\omega\|^{2p})\,\mathrm{d}\hat\rho_t^N \right]$ is bounded for $t \in [0, T]$ and the bound does not depend on $N$. Also, we know that

$$\|M_\beta[\hat\mu_t^N \rho_t]\|^{2p} \lesssim 1 + \int (\|\theta\|^{2p} + \|\omega\|^{2p})\mathrm{d}\hat\rho_\tau^N,$$

which implies that

$$\mathbb{E}\left[ \|M_\beta[\hat\mu_t^N]\|^{2p} \right] \lesssim 1 + \mathbb{E}\left[ \int (\|\theta\|^{2p} + \|\omega\|^{2p})\mathrm{d}\hat\rho_\tau^N \right].$$

So $\mathbb{E}\left[ \|M_\beta[\hat\mu_t^N]\|^{2p} \right]$ is bounded for $t \in [0, T]$ and the bound does not depend on $N$. $\qquad\square$

Now we treat $\left( \Theta^{(i,N)}, \Omega^{(i,N)} \right)$ as a random variable defined on $(\Omega, \mathcal{F}, \mathbb{P})$ and taking values in $\mathcal{C}\left( [0,T]; \mathbb{R}^D \times \mathbb{R}^D \right)$. Then $\hat\rho^N = \frac{1}{N} \sum_{i=1}^N \delta_{(\Theta^{(i,N)}, \Omega^{(i,N)})}$ is a random measure. Let us denote $\mathcal{L}(\hat\rho^N) := Law(\rho^N) \in \mathcal{P}\left( \mathcal{P}\left( \mathcal{C}([0,T], \mathcal{R}^D) \times \mathcal{C}\left( [0,T], \mathcal{R}^D \right) \right) \right)$ as a sequence of probability distributions. We can prove that $\left\{ \mathcal{L}(\rho^N) \right\}_{N \ge 2}$ is tight. Next, we use the Aldous criteria ((Bass, 2011), Section 34.3), which could prove the tightness of a sequence of distributions.

**Theorem C.2.** *Let $\mathcal{J}$ satisfy Assumptions 4.1 and $\rho_0 \in \mathcal{P}_4(\mathbb{R}^D \times \mathbb{R}^D)$. For any $N \ge 2$, we assume that $\left\{ \left( \Theta_t^{(i,N)}, \Omega_t^{(i,N)} \right)_{t \in [0,T]} \right\}_{i=1}^N$ is a unique solution to Equation (8) with $\rho_0^{\otimes N}$-distributed initial data $\left\{ \left( \Theta_0^{(i,N)}, \Omega_0^{(i,N)} \right) \right\}_{i=1}^N$. Then the sequence $\left\{ \mathcal{L}(\hat\rho^N) \right\}_{N \ge 2}$ is tight in $\mathcal{P}\left( \mathcal{P}\left( \mathcal{C}([0,T], \mathcal{R}^D) \times \mathcal{C}\left( [0,T], \mathcal{R}^D \right) \right) \right)$.*

*Proof.* Because of the exchangeability of the particle system, we only prove the $\left\{ \mathcal{L}\left( \Theta_t^{1,N}, \Omega_t^{1,N} \right) \right\}_{N \ge 2}$ is tight. It is sufficient to justify two conditions in Aldous criteria.

**Condtion 1:** For any $\epsilon > 0$, there exist a compact subset $U_\epsilon := \left\{ (\theta, \omega) : \|\theta\|^2 + \|\omega\|^2 \leq \frac{K}{\epsilon} \right\}$ such that by Markov's inequality

$$\mathcal{L}\left(\Theta_t^{1,N}, \Omega_t^{1,N}\right)((U_\epsilon)^c) = \mathbb{P}\left(\left\|\Theta_t^{1,N}\right\| + \left\|\Omega_t^{1,N}\right\| > \frac{K}{\epsilon}\right) \leq \frac{\epsilon \mathbb{E}[\|\Theta_t^{1,N}\| + \|\Omega_t^{1,N}\|]}{K} \leq \epsilon, \quad \forall N \geq 2,$$

where we have used Lemma C.1 in the last inequality. This means that for each $t \in [0, T]$, the sequence $\left\{ \mathcal{L}\left(\Theta_t^{1,N}, \Omega_t^{1,N}\right) \right\}$ is tight.

**Condition 2:** We have to show, for any $\epsilon, \eta > 0$, there exist $\delta_0 > 0$ and $n_0 \in \mathbb{N}$ such that for all $N \geq n_0$. Let $\tilde{\tau}$ be a $\sigma((\Theta_s^{1,N}, \Omega_s^{1,N}); s \in [0, T])$-stopping time with discrete values such that $\tilde{\tau} + \delta_0 \leq T$, it holds that

$$\sup_{\delta \in [0, \delta_0]} \mathbb{P}\left(\left\|\Theta_{t+\delta}^{1,N} - \Theta_t^{1,N}\right\| \geq \eta\right) \leq \epsilon$$

and

$$\sup_{\delta \in [0, \delta_0]} \mathbb{P}\left(\left\|\Omega_{t+\delta}^{1,N} - \Omega_t^{1,N}\right\| \geq \eta\right) \leq \epsilon.$$

Recalling Equation (1), we have

$$\Theta_{\tilde{\tau}+\delta}^{1,N} - \Theta_{\tilde{\tau}}^{1,N} = \int_{\tilde{\tau}}^{\tilde{\tau}+\delta} \Omega_s^{1,N} \mathrm{d}s - \gamma_1 \int_{\tilde{\tau}}^{\tilde{\tau}+\delta} (\Theta_s^{1,N} - \mathcal{M}_\beta[\hat{\mu}_s^N]) \mathrm{d}s + \int_{\tilde{\tau}}^{\tilde{\tau}+\delta} \sigma(s) \mathrm{d}W_{\theta,t},$$

$$\Omega_{\tilde{\tau}+\delta}^{1,N} - \Omega_{\tilde{\tau}}^{1,N} = -m \int_{\tilde{\tau}}^{\tilde{\tau}+\delta} (\Theta_s^{1,N} - \mathcal{M}_\beta[\hat{\mu}_s^N]) \mathrm{d}s - \gamma_2 \int_{\tilde{\tau}}^{\tilde{\tau}+\delta} \Omega_s^{1,N} \mathrm{d}s + \sqrt{m} \int_{\tilde{\tau}}^{\tilde{\tau}+\delta} \sigma(s) \mathrm{d}W_{\omega,t}.$$

From Theorem 2.1 in (Huang & Qiu, 2022), we have

$$\mathbb{E}\left[\left|\int_{\tilde{\tau}}^{\tilde{\tau}+\delta} (\Theta_s^{1,N} - \mathcal{M}_\beta[\hat{\mu}_s^N]) \mathrm{d}s\right|^2\right] \leq 2TK\delta.$$

Furthermore, we apply Itô's Isometry

$$\mathbb{E}\left[\int_{\tilde{\tau}}^{\tilde{\tau}+\delta} \sigma(s) \mathrm{d}W_{\theta,s}\right], \mathbb{E}\left[\int_{\tilde{\tau}}^{\tilde{\tau}+\delta} \sigma(s) \mathrm{d}W_{\omega,s}\right] \leq \left(\mathbb{E}\left[\int_{\tilde{\tau}}^{\tilde{\tau}+\delta} \sigma(s)^2 \mathrm{d}s\right]\right)^{\frac{1}{2}} \leq \bar{\sigma}^2 \delta^{\frac{1}{2}} T^{\frac{1}{2}}.$$

Combining the above estimation, one has

$$\mathbb{E}\left[\left|\Theta_{\tilde{\tau}+\delta}^{1,N} - \Theta_{\tilde{\tau}}^{1,N}\right|^2 + \left|\Omega_{\tilde{\tau}+\delta}^{1,N} - \Omega_{\tilde{\tau}}^{1,N}\right|^2\right] \lesssim \sqrt{\delta}.$$

Hence, for any $\epsilon, \eta > 0$, there exist $\delta_0 > 0$ such that for all $N > 2$ it holds that

$$\sup_{\delta \in [0, \delta_0]} \mathbb{P}\left(\left\|\Theta_{t+\delta}^{1,N} - \Theta_t^{1,N}\right\|^2 \geq \eta\right), \sup_{\delta \in [0, \delta_0]} \mathbb{P}\left(\left\|\Omega_{t+\delta}^{1,N} - \Omega_t^{1,N}\right\|^2 \geq \eta\right)$$

$$\leq \sup_{\delta \in [0, \delta_0]} \mathbb{P}\left(\left\|\Theta_{t+\delta}^{1,N} - \Theta_t^{1,N}\right\|^2 + \|\Omega_{t+\delta}^{1,N} - \Omega_t^{1,N}\|^2 \geq \eta\right)$$

$$\leq \sup_{\delta \in [0, \delta_0]} \frac{\mathbb{E}\left[\left|\Theta_{\tilde{\tau}+\delta}^{1,N} - \Theta_{\tilde{\tau}}^{1,N}\right|^2 + \left|\Omega_{\tilde{\tau}+\delta}^{1,N} - \Omega_{\tilde{\tau}}^{1,N}\right|^2\right]}{\eta} \leq \epsilon.$$

$\square$

By Skorokhod's lemma (see (Billingsley, 2013)) and Lemma C.3, we may find a common probability space $(\Omega, \mathcal{F}, \mathbb{P})$ on which the process $\{\hat{\rho}^N\}_{N \in \mathbb{N}}$ converges to some process $\rho$ as a random variable valued in $\mathcal{P}\left(\mathcal{C}\left([0, T]; \mathbb{R}^D\right) \times \mathcal{C}\left([0, T]; \mathbb{R}^D\right)\right)$ almost surely. In particular, we have that $\forall t \in [0, T]$ and $\phi \in \mathcal{C}_b(\mathbb{R}^D \times \mathbb{R}^D)$,

$$\lim_{N \to \infty} \left|\langle \phi, \hat{\rho}_t^N - \rho_t \rangle\right| + \left|\mathcal{M}_\beta[\hat{\mu}_t^N] - \mathcal{M}_\beta[\mu_t]\right| = 0 \quad \text{a.s.,} \tag{11}$$

and we can have a direct result

$$\lim_{N \to \infty} \mathbb{E}\left[\left|\langle \phi, \hat{\rho}_t^N - \rho_t \rangle\right| + \left|\mathcal{M}_\beta[\hat{\mu}_t^N] - \mathcal{M}_\beta[\mu_t]\right|\right] = 0. \tag{12}$$

**Lemma C.3.** *1. There exist a subsequence of $\{\hat{\rho}^N\}_{N \geq 2}$ and a random measure $\rho : \Omega \to \mathcal{P}\left(\mathcal{C}\left([0, T]; \mathbb{R}^D\right) \times \mathcal{C}\left([0, T]; \mathbb{R}^D\right)\right)$ such that $\hat{\rho}^N \rightharpoonup \rho$ in law as $N \to \infty$ which is equivalently to say $\mathcal{L}\left(\hat{\rho}^N\right)$ converges weakly to $\mathcal{L}(\rho)$ in $\mathcal{P}\left(\mathcal{P}\left(\mathcal{C}\left([0, T]; \mathbb{R}^D\right) \times \mathcal{C}\left([0, T]; \mathbb{R}^D\right)\right)\right);$*

*2. For the subsequence in 1, the time marginal $\hat{\rho}_t^N$ of $\hat{\rho}^N$, as $\mathcal{P}\left(\mathbb{R}^D \times \mathbb{R}^D\right)$ valued random measure converges in law to $\rho_t \in \mathcal{P}(\mathbb{R}^D \times \mathbb{R}^D)$, the time marginal of $\rho$. Namely $\mathcal{L}(\hat{\rho}_t^N)$ converges weakly to $\mathcal{L}(\rho_t)$ in $\mathcal{P}\left(\mathcal{P}\left(\mathbb{R}^D \times \mathbb{R}^D\right)\right).$*

**Definition C.4.** Fix $\phi \in \mathcal{C}_c^2\left(\mathbb{R}^D \times \mathbb{R}^D\right)$, define functional $\mathcal{F}_\phi : \mathcal{P}\left(\mathcal{C}\left([0, T]; \mathbb{R}^D\right) \times \mathcal{C}\left([0, T]; \mathbb{R}^D\right)\right) \to \mathbb{R}$:

$$\begin{aligned}
F_\phi(\rho_t) := &\langle \phi(\theta_t, \omega_t), \rho(\mathrm{d}\theta, \mathrm{d}\omega) \rangle - \langle \varphi(\theta_0, \omega_0), \rho(\mathrm{d}\theta, \mathrm{d}\omega) \rangle \\
&- \int_0^t \langle (\omega_s - \gamma_1(\theta_s - \mathcal{M}_\beta[\mu_s])) \cdot \nabla_\theta \phi, \rho(\mathrm{d}\theta, \mathrm{d}\omega) \rangle \, \mathrm{d}s \\
&- \int_0^t \langle (-m(\theta_s - \mathcal{M}_\beta[\mu_s]) - \gamma_2 \omega_s) \cdot \nabla_\omega \phi, \rho(\mathrm{d}\theta, \mathrm{d}\omega) \rangle \, \mathrm{d}s \\
&- \frac{(m+1)D}{2} \int_0^t \sigma^2(s) \mathrm{d}s
\end{aligned} \tag{13}$$

for all $\rho \in \mathcal{P}\left(\mathcal{C}\left([0, T]; \mathbb{R}^D\right) \times \mathcal{C}\left([0, T]; \mathbb{R}^D\right)\right)$ and $\theta, \omega \in \mathcal{C}\left([0, T]; \mathbb{R}^D\right)$, where $\mu(\theta) = \int_{\mathbb{R}^D} \rho(\theta, \mathrm{d}\omega)$.

**Lemma C.5.** *Let $\mathcal{J}$ satisfy Assumption 4.1 and $\rho_0 \in \mathcal{P}_4(\mathbb{R}^D \times \mathbb{R}^D)$. For all $N \geq 2$, assume that $\left\{\left(\Theta_t^{(i,N)}, \Omega_t^{(i,N)}\right)_{t \in [0,T]}\right\}_{i=1}^N$ is the unique solution to Equation (8) with $\rho_0^{\otimes N}$-distributed initial data $\left\{\left(\Theta_0^{(i,N)}, \Omega_0^{(i,N)}\right)\right\}_{i=1}^N$. There exists a constant $C > 0$, such that*

$$\mathbb{E}\left[\left|F_\phi(\hat{\rho}^N)\right|^2\right] \leq \frac{C}{N},$$

*where $\hat{\rho}^N = \frac{1}{N} \sum_{i=1}^N \delta_{(\Theta^{(i,N)}, \Omega^{(i,N)})}$ is the empirical measure.*

*Proof.* Using the definition of $F_\phi$, one has

$$\begin{aligned}
F_\phi(\hat{\rho}^N) = &\frac{1}{N} \sum_{i=1}^N \phi(\Theta_t^{(i,N)}, \Omega_t^{(i,N)}) - \frac{1}{N} \sum_{i=1}^N \phi(\Theta_0^{(i,N)}, \Omega_0^{(i,N)}) \\
&- \frac{1}{N} \sum_{i=1}^N \int_0^t \left(\Omega_s^{(i,N)} - \gamma_1 \left(\Theta_0^{(i,N)} - \mathcal{M}_\beta[\hat{\mu}_s^N]\right)\right) \cdot \nabla_\theta \phi(\Theta_s^{(i,N)}, \Omega_s^{(i,N)}) \mathrm{d}s \\
&- \frac{1}{N} \sum_{i=1}^N \int_0^t \left(-m\left(\Theta_s^{(i,N)} - \mathcal{M}_\beta[\hat{\mu}_s^N]\right) - \gamma_2 \Omega_s^{(i,N)}\right) \cdot \nabla_\omega \phi(\Theta_s^{(i,N)}, \Omega_s^{(i,N)}) \mathrm{d}s \\
&- \frac{(m+1)D}{2} \int_0^t \sigma^2(s) \mathrm{d}s.
\end{aligned}$$

One the other hand, the Itô-Doeblin formula gives

$$
\begin{aligned}
\phi\left(\Theta_t^{(i,N)}, \Omega_t^{(i,N)}\right) - \phi\left(\Theta_0^{(i,N)}, \Omega_0^{(i,N)}\right) &= \int_0^t \left(\Omega_s^{(i,N)} - \gamma_1\left(\Theta_s^{(i,N)} - \mathcal{M}_\beta[\hat\mu_s^N]\right)\right) \cdot \nabla_\theta \phi(\Theta_s^{(i,N)}, \Omega_s^{(i,N)}) \mathrm{d}s \\
&\quad + \int_0^t \left(-m\left(\Theta_s^{(i,N)} - \mathcal{M}_\beta[\hat\mu_s^N]\right) - \gamma_2 \Omega_s^{(i,N)}\right) \cdot \nabla_\omega \phi(\Theta_s^{(i,N)}, \Omega_s^{(i,N)}) \mathrm{d}s \\
&\quad + \int_0^t \sigma(s)\mathrm{d}W_{\theta,s}^i + \sqrt{m} \int_0^t \sigma(s)\mathrm{d}W_{\omega,s}^i \\
&\quad + \frac{(m+1)D}{2} \int_0^t \sigma^2(s)\mathrm{d}s.
\end{aligned}
$$

Then one gets

$$
F_\phi(\hat\rho^N) = \frac{1}{N} \sum_{i=1}^N \left(\int_0^t \sigma(s)\mathrm{d}W_{\theta,s}^i + \sqrt{m} \int_0^t \sigma(s)\mathrm{d}W_{\omega,s}^i\right).
$$

Finally, we can compute

$$
\begin{aligned}
\mathbb{E}\left[|F_\phi(\hat\rho^N)|^2\right] &= \frac{1}{N^2} \sum_{i=1}^N \mathbb{E}\left[\left|\int_0^t \sigma(s)\mathrm{d}W_{\theta,s}^i + \sqrt{m} \int_0^t \sigma(s)\mathrm{d}W_{\omega,s}^i\right|\right]^2 \\
&\leq T \frac{\overline\sigma^2(m+1)}{N^2},
\end{aligned}
$$

where we use the assumption that the $\sigma(t)$ we choose has an upper bound. $\qquad \square$

*Proof of Theorem 4.4.* Now suppose that we have a convergent subsequence of $\{\hat\rho^N\}_{N\in\mathbb{N}}$, which is denoted by the sequence itself for simplicity and has $\rho_t$ as the limit. We want to prove that $\rho$ is a solution of (3). For any $\phi \in \mathcal{C}_c^2\left(\mathbb{R}^D \times \mathbb{R}^D\right)$, using the convergence result in (12), we have

$$
\lim_{N\to\infty} \mathbb{E}\left[\left|\langle \phi(\theta,\omega), \hat\rho_t^N(\mathrm{d}\theta, \mathrm{d}\omega)\rangle - \langle \phi(\theta,\omega), \rho_t(\mathrm{d}\theta, \mathrm{d}\omega)\rangle\right|\right] = 0,
$$

and

$$
\lim_{N\to\infty} \mathbb{E}\left[\left|\langle \phi(\theta,\omega), \hat\rho_0^N(\mathrm{d}\theta, \mathrm{d}\omega)\rangle - \langle \phi(\theta,\omega), \rho_0(\mathrm{d}\theta, \mathrm{d}\omega)\rangle\right|\right] = 0.
$$

Further, we notice that

$$
\begin{aligned}
&\left\| \int_0^t \langle (\theta - \mathcal{M}_\beta[\hat\mu_s^N]) \cdot \nabla_\theta \phi, \hat\rho_s^N(\mathrm{d}\theta, \mathrm{d}\omega)\rangle \mathrm{d}s - \int_0^t \langle (\theta - \mathcal{M}_\beta[\mu_s]) \cdot \nabla_\theta \phi, \rho_s(\mathrm{d}\theta, \mathrm{d}\omega)\rangle \mathrm{d}s \right\| \\
&\leq \int_0^t \left\| \langle (\theta - \mathcal{M}_\beta[\hat\mu_s^N]) \cdot \nabla_\theta \phi, \hat\rho_s^N(\mathrm{d}\theta, \mathrm{d}\omega) - \rho_s(\mathrm{d}\theta, \mathrm{d}\omega)\rangle \right\| \mathrm{d}s \\
&\quad + \int_0^t \left\| \langle (\mathcal{M}_\beta[\mu_s] - \mathcal{M}_\beta[\hat\mu_s^N]) \cdot \nabla_\theta \phi, \rho_s(\mathrm{d}\theta, \mathrm{d}\omega)\rangle \right\| \mathrm{d}s \\
&:= \int_0^t \left\| I_1^N(s) \right\| \mathrm{d}s + \int_0^t \left\| I_2^N(s) \right\| \mathrm{d}s.
\end{aligned}
$$

For $\left\| I_1^N(s) \right\|$, we have

$$
\begin{aligned}
\mathbb{E}\left[\left\| I_1^N(s) \right\|\right] &= \mathbb{E}\left[\left\| \langle \theta \cdot \nabla_\theta \phi, \rho_s^N(\mathrm{d}\theta, \mathrm{d}\omega) - \rho_s(\mathrm{d}\theta, \mathrm{d}\omega)\rangle \right\|\right] + \mathbb{E}\left[\left\| \langle (\mathcal{M}_\beta[\mu_s^N]) \cdot \nabla_\theta \phi, \rho_s^N(\mathrm{d}\theta, \mathrm{d}\omega) - \rho_s(\mathrm{d}\theta, \mathrm{d}\omega)\rangle \right\|\right] \\
&\leq \mathbb{E}\left[\left\| \langle \theta \cdot \nabla_\theta \phi, \rho_s^N(\mathrm{d}\theta, \mathrm{d}\omega) - \rho_s(\mathrm{d}\theta, \mathrm{d}\omega)\rangle \right\|\right] + K^{\frac{1}{2}} \mathbb{E}\left[\left\| \langle \nabla_\theta \phi, \rho_s^N(\mathrm{d}\theta, \mathrm{d}\omega) - \rho_s(\mathrm{d}\theta, \mathrm{d}\omega)\rangle \right\|\right].
\end{aligned}
\tag{14}
$$

Since $\phi$ has a compact support, applying (12) leads to $\lim_{N\to\infty} \mathbb{E}\left[\left\| I_1^N(s) \right\|\right] = 0$. Moreover, by the uniform boundedness of $\mathbb{E}\left[I_1^N(s)\right] = 0$ and applying the domained convergence theorem implies

$$
\lim_{N\to\infty} \int_0^t \mathbb{E}\left[\left\| I_1^N(s) \right\|\right] \mathrm{d}s = 0.
$$

As for $I_2^N$, we know that

$$\left\| \langle (\mathcal{M}_\beta(\mu_s) - \mathcal{M}_\beta(\hat{\mu}_s^N)) \cdot \nabla\phi(x), \mu_s(dx) \rangle \right\| \le \|\nabla\phi\|_\infty \|\mathcal{M}_\beta(\mu_s) - \mathcal{M}_\beta(\hat{\mu}_s^N)\|.$$

Hence by Equation (12), we have $\lim_{N\to\infty} \mathbb{E}\left[\|I_2^N(s)\|\right] = 0$. Again, by the dominated convergence theorem, we have

$$\lim_{N\to\infty} \int_0^t \mathbb{E}\left[\|I_2^N(s)\|\right] ds = 0.$$

Thus we can get the boundedness

$$\lim_{N\to\infty} \mathbb{E}\left[\left| \int_0^t \langle (\theta - \mathcal{M}_\beta[\hat{\mu}_s^N]) \cdot \nabla_\theta\phi, \rho_s^N(d\theta, d\omega) \rangle ds - \int_0^t \langle (\theta - \mathcal{M}_\beta[\hat{\mu}_s]) \cdot \nabla_\theta\phi, \rho_s(d\theta, d\omega) \rangle ds \right|\right] = 0.$$

Similarly, we can also have

$$\lim_{N\to\infty} \mathbb{E}\left[\left| \int_0^t \langle (\theta - \mathcal{M}_\beta[\hat{\mu}_s^N]) \cdot \nabla_\omega\phi, \rho_s^N(d\theta, d\omega) \rangle ds - \int_0^t \langle (\theta - \mathcal{M}_\beta[\hat{\mu}_s]) \cdot \nabla_\omega\phi, \rho_s(d\theta, d\omega) \rangle ds \right|\right] = 0.$$

Combining the above results, we get

$$\mathbb{E}\left[\left|F_\phi(\hat{\rho}^N) - F_\phi(\rho)\right|\right] = 0,$$

which is a direct result of (12) and the dominated convergence theorem. We can deduce

$$\begin{aligned} \mathbb{E}\left[|F_\phi(\rho)|\right] \le & \mathbb{E}\left[\left|F_\phi(\hat{\rho}^N) - F_\phi(\rho)\right|\right] + \mathbb{E}\left[\left|F_\phi(\hat{\rho}^N)\right|\right] \\ \le & \mathbb{E}\left[\left|F_\phi(\hat{\rho}^N) - F_\phi(\rho)\right|\right] + \sqrt{\frac{C}{N}} \to 0, \quad \text{as} \quad N \to \infty. \end{aligned}$$

Thus $F_\phi(\rho_t) = 0$ almost surely, which implies that $\rho_t$ is a solution to the corresponding Fokker-Planck equation. Combined with the uniqueness of the solution, proved in Lemma C.7, we complete the proof. $\square$

**Lemma C.6.** $\forall T > 0$, let $f_t \in \mathcal{C}\left([0,T]; \mathbb{R}^D\right)$ and $\rho_0 \in \mathcal{P}_2(\mathbb{R}^D \times \mathbb{R}^D)$. The following linear PDE

$$\partial_t \rho_t = -\nabla_\theta\left((\omega - \gamma_1(\theta - f_t))\rho_t\right) + \nabla_\omega\left(m(\theta - f_t) + \gamma_2\omega\rho_t\right) + \frac{\sigma(t)^2 m}{2}\Delta_\omega\rho_t + \frac{\sigma(t)^2}{2}\Delta_\theta\rho_t$$

has a unique solution $\rho_t \in \mathcal{C}\left([0,T]; \mathcal{P}_2(\mathbb{R}^D \times \mathbb{R}^D)\right)$.

*Proof.* The existence is obvious, which can be obtained as the law of the solution to the associated linear SDE. To show the uniqueness, let us fix $t_0 \in [0,T]$ and $\phi \in \mathcal{C}_c^\infty\left(\mathbb{R}^D \times \mathbb{R}^D\right)$, we then can solve the following backward PDE:

$$\partial_t h_t = -(\omega - \gamma_1(\theta - f_t)) \cdot \nabla_\theta h_t + (m(\theta - f_t) + \gamma_2\omega) \cdot \nabla_\omega h_t - \frac{\sigma(t)^2}{2}\left(m\Delta_\omega h_t + \Delta_\theta h_t\right),$$

where $(t, \theta, \omega) \in [0, t_0] \times \mathbb{R}^D \times \mathbb{R}^D$ with terminal condition $h_{t_0} = \phi$. It has a classical solution:

$$h_t = \mathbb{E}\left[\phi(\Theta_{t_0}^{t,(\theta,\omega)}, \Omega_{t_0}^{t,(\theta,\omega)})\right],$$

where $\left(\Theta_s^{t,(\theta,\omega)}, \Omega_s^{t,(\theta,\omega)}\right)_{0 \le t \le s \le t_0}$ is the strong solution to the following linear SDE:

$$\begin{aligned} d\Theta_s^{t,(\theta,\omega)} &= \left(\Omega_s^{t,(\theta,\omega)} - \gamma_1(\Theta_s^{t,(\theta,\omega)} - f_s)\right) ds + \sigma(s) dW_{\theta,s}, \\ d\Omega_s^{t,(\theta,\omega)} &= -\left(m(\Theta_s^{t,(\theta,\omega)} - f_s) + \gamma_2\Omega_s^{t,(\theta,\omega)}\right) ds + \sigma(s)\sqrt{m} dW_{\omega,s} \end{aligned}$$

with terminal condition $\left(\Theta_t^{t,(\theta,\omega)}, \Omega_t^{t,(\theta,\omega)}\right) = (\theta, \omega)$.

Now, suppose $\rho^1$ and $\rho^2$ are two weak solutions of the PDE with the same initial condition $\rho_0^1 = \rho_0^2$. Let $\delta\rho = \rho^1 - \rho^2$. To show uniqueness, we need to demonstrate that $\delta\rho_t = 0$ for all $t \in [0, T]$.

Using the backward PDE solution $h_t$ as a test function, we have:

$$\langle h_{t_0}, \delta\rho_{t_0} \rangle = \int_0^{t_0} \langle \partial_s h_s, \delta\rho_s \rangle + \langle h_s, \partial_s \delta\rho_s \rangle \mathrm{d}s.$$

By substituting the equation for $\partial_s h_s$ from the backward PDE and integrating by parts, we get

$$\langle h_{t_0}, \delta\rho_{t_0} \rangle = 0.$$

The arbitrariness of $\psi \in \mathcal{C}_c^\infty(\mathbb{R}^D \times \mathbb{R}^D)$ implies $\delta\rho_{t_0} = 0$, and thus $\rho^1 = \rho^2$. $\qquad\square$

**Lemma C.7.** *Assume that $\rho^1, \rho^2 \in \mathcal{C}\left([0,T]; \mathcal{P}_2\left(\mathbb{R}^D \times \mathbb{R}^D\right)\right)$ are two weak solutions to Equation* (3) *with the same initial data $\rho_0$. Then it holds that*

$$\sup_{t \in [0,T]} W_2\left(\rho_t^1, \rho_t^2\right) = 0,$$

*where $W_2$ is the 2-Wasserstein distance.*

*Proof.* Given $\rho^1$ and $\rho^2$, let us first consider the following two coupled linear SDEs:

$$\mathrm{d}\bar{\Theta}_t^i = \left(\bar{\Omega}_t^i - \gamma_1\left(\bar{\Theta}_t^i - \mathcal{M}_\beta[\mu_t^i]\right)\right)\mathrm{d}t + \sigma(t)\mathrm{d}W_{\theta,t},$$
$$\mathrm{d}\bar{\Omega}_t^i = -\left(m\left(\bar{\Theta}_t^i - \mathcal{M}_\beta[\mu_t^i]\right) + \gamma_2\bar{\Omega}_t^i\right)\mathrm{d}s + \sigma(t)\sqrt{m}\mathrm{d}W_{\omega,t},$$

for $i = 1, 2$, where $(\bar{\Theta}_t^i, \bar{\Omega}_t^i)$ are driven by independent Brownian motions $W_{\theta,s}$ and $W_{\omega,s}$, with the same initial condition $(\Theta_0^{i,t}, \Omega_0^{i,t}) \sim \rho_0$.

Let $\tilde{\rho}_t^i$ denote the law of $(\Theta_t^i, \Omega_t^i)$ for $i = 1, 2$. By construction, the laws $\tilde{\rho}_t^i$ satisfy the same Fokker-Planck equation:

$$\partial_t \tilde{\rho}_t^i = -\nabla_\theta \cdot \left[\left(\omega - \gamma_1(\theta - \mathcal{M}_\beta[\mu_t^i])\right)\tilde{\rho}_t^i\right] + \nabla_\omega \cdot \left[\left(m(\theta - \mathcal{M}_\beta[\mu_t^i]) + \gamma_2\omega\right)\tilde{\rho}_t^i\right]$$
$$+ \frac{\sigma(t)^2 m}{2}\Delta_\omega \tilde{\rho}_t^i + \frac{\sigma(t)^2}{2}\Delta_\theta \tilde{\rho}_t^i,$$

in the weak sense, with initial condition $\tilde{\rho}_0^i = \rho_0$. Since both $\rho_t^i$ solve this Fokker-Planck equation and we assumed $\rho_0^1 = \rho_0^2 = \rho_0$, by the uniqueness of solutions to this PDE in the last lemma, we have $\tilde{\rho}_t^i = \rho_t$ for $i = 1, 2$. As a result, $(\Theta_t^{1,t}, \Omega_t^{1,t})$ and $(\Theta_t^{2,t}, \Omega_t^{2,t})$ both solve Equation (2). By Theorem 4.3, we have

$$\sup_{t \in [0,T]} \mathbb{E}\left[\left|\Theta_t^{1,t} - \Theta_t^{2,t}\right|^2 + \left|\Omega_t^{1,t} - \Omega_t^{2,t}\right|^2\right] = 0.$$

This implies:

$$\sup_{t \in [0,T]} W_2\left(\tilde{\rho}_t^1, \tilde{\rho}_t^2\right) \leq \sup_{t \in [0,T]} \mathbb{E}\left[\left|\bar{\Theta}_t^{1,t} - \bar{\Theta}_t^{2,t}\right|^2 + \left|\bar{\Omega}_t^{1,t} - \bar{\Omega}_t^{2,t}\right|^2\right] = 0.$$

$\qquad\square$

## D. Global Convergence in the Mean Field Law

**Lemma D.1.** *Let $E[\rho_t]$ be the energy functional defined in* (4). *Under Assumption* 4.1,

$$\frac{\mathrm{d}}{\mathrm{d}t}E[\rho_t] \leq -\gamma E[\rho_t] + \lambda\sqrt{E[\rho_t]}\|\mathcal{M}_\beta[\mu_t] - \tilde{\theta}\| + \frac{\sigma^2(t)D(m+1)}{2},$$

*where $\gamma = \min\{\gamma_1, \gamma_2\}$ and $\lambda = \max\{m, \gamma_1\}$ are positive numbers.*

*Proof.* From the definition of weak solution of Fokker-Planck equation (3), we define $\phi(\theta, \omega) = \frac{1}{2}\|\theta - \tilde{\theta}\|^2 + \frac{1}{2m}\|\omega\|^2$, then

$$
\begin{aligned}
\frac{\mathrm{d}}{\mathrm{d}t} E[\rho_t] &= \frac{\mathrm{d}}{\mathrm{d}t} \int \phi(\theta, \omega) \mathrm{d}\rho_t \\
&= \int \langle \omega - \gamma_1(\theta - \mathcal{M}_\beta[\mu_t]), \theta - \tilde{\theta} \rangle \mathrm{d}\rho_t + \int \langle -m(\theta - \mathcal{M}_\beta[\mu_t]) - \gamma_2 \omega, \frac{1}{m}\omega \rangle \mathrm{d}\rho_t + \frac{\sigma^2 D(m+1)}{2} \\
&= \int \langle \omega, \mathcal{M}_\beta[\mu_t] - \tilde{\theta} \rangle \mathrm{d}\rho_t - \gamma_1 \int \langle \theta - \mathcal{M}_\beta[\mu_t], \theta - \tilde{\theta} \rangle \mathrm{d}\rho_t - \frac{\gamma_2}{m} \int \langle \omega, \omega \rangle \mathrm{d}\rho_t + \frac{\sigma^2 D(m+1)}{2} \\
&= \int \langle \omega, \mathcal{M}_\beta[\mu_t] - \tilde{\theta} \rangle \mathrm{d}\rho_t - \gamma_1 \int \langle \theta - \tilde{\theta}, \theta - \tilde{\theta} \rangle \mathrm{d}\rho_t \\
&\quad - \gamma_1 \int \langle \tilde{\theta} - \mathcal{M}_\beta[\mu_t], \theta - \tilde{\theta} \rangle \mathrm{d}\rho_t - \frac{\gamma_2}{m} \int \langle \omega, \omega \rangle \mathrm{d}\rho_t + \frac{\sigma^2 D(m+1)}{2} \\
&\leq \|\omega\|\|\mathcal{M}_\beta[\mu_t] - \tilde{\theta}\| + \gamma_1 \|\theta - \tilde{\theta}\|\|\mathcal{M}_\beta[\mu_t] - \tilde{\theta}\| - \gamma_1 \|\theta - \tilde{\theta}\|^2 - \frac{\gamma_2}{m}\|\omega\|^2 + \frac{\sigma^2 D(m+1)}{2} \\
&\leq -\gamma E[\rho_t] + \lambda \sqrt{E[\rho_t]} \|\mathcal{M}_\beta[\mu_t] - \tilde{\theta}\| + \frac{\sigma^2 D(m+1)}{2},
\end{aligned}
\tag{15}
$$

where in the last equility we take $\gamma = \min\{\gamma_1, \gamma_2\}$ and $\lambda = \max\{m, \gamma_1\}$. $\qquad \square$

Next, we will show that $\|\mathcal{M}_\beta[\mu_t] - \tilde{\theta}\|$ can be bounded by a suitable scalar multiple of $\sqrt{E[\rho_t]}$, we can apply Gronwall's inequality to bound the energy function.

**Lemma D.2.** *Under Assumption 4.1, $\forall r > 0$, we define $J_r := \sup_{\theta \in B_{\theta,r}(\tilde{\theta})} \mathcal{J}(\theta)$. Then $\forall r \in [0, R_0]$ and $q > 0$ such that $(q + J_r - \underline{J})^\mu \leq \delta_J$, we have*

$$
\|\mathcal{M}_\beta[\mu] - \tilde{\theta}\| \leq \frac{(q + J_r - \underline{J})^\mu}{\eta} + \frac{\exp(-\beta q)}{\rho(B_{\theta,r}(\tilde{\theta}))} \int \|\theta - \tilde{\theta}\| \mathrm{d}\rho(\theta, \omega).
$$

*Proof.* Let $\tilde{r} = \frac{(q + J_r - \underline{J})^\mu}{\eta} \geq \frac{(J_r - \underline{J})^\mu}{\eta} \geq r$, we have

$$
\begin{aligned}
\|\mathcal{M}_\beta[\mu] - \tilde{\theta}\| &\leq \int_{B_{\theta,\tilde{r}}(\tilde{\theta})} \|\theta - \tilde{\theta}\| \frac{w_\beta(\theta)}{\|w_\beta(\theta)\|_{L^1(\rho)}} \mathrm{d}\rho + \int_{B^c_{\theta,\tilde{r}}(\tilde{\theta})} \|\theta - \tilde{\theta}\| \frac{w_\beta(\theta)}{\|w_\beta(\theta)\|_{L^1(\rho)}} \mathrm{d}\rho \\
&\leq \tilde{r} + \int_{B^c_{\theta,\tilde{r}}(\tilde{\theta})} \|\theta - \tilde{\theta}\| \frac{w_\beta(\theta)}{\|w_\beta(\theta)\|_{L^1(\rho)}} \mathrm{d}\rho.
\end{aligned}
\tag{16}
$$

By Markov's inequality, we have $\|w_\beta\|_{L^1(\rho)} \geq a\rho(\{(\theta, \omega) : \exp(-\beta \mathcal{J}(\theta) \geq a)\})$. By choosing $a = \exp(-\beta J_r)$, we have

$$
\begin{aligned}
\|w_\beta\|_{L^1(\rho)} &\geq \exp(-\beta J_r)\rho\left(\{(\theta, \omega) : \exp(-\beta \mathcal{J}(\theta) \geq \exp(-\beta J_r))\}\right) \\
&= \exp(-\beta J_r)\rho\left(\{(\theta, \omega) : J(\theta) \leq J_r\}\right) \\
&\geq \exp(-\beta J_r)\rho(B_{\theta,r}(\tilde{\theta})),
\end{aligned}
$$

where the second inequality comes from the definition of $J_r$. Thus for the second term in (16), we obtain

$$
\begin{aligned}
\int_{B^c_{\theta,\tilde{r}}(\tilde{\theta})} \|\theta - \tilde{\theta}\| \frac{w_\beta(\theta)}{\|w_\beta(\theta)\|_{L^1(\rho)}} \mathrm{d}\rho &\leq \frac{1}{\exp(-\beta J_r)\rho(B_{\theta,r}(\tilde{\theta}))} \int_{B^c_{\theta,\tilde{r}}(\tilde{\theta})} \|\theta - \tilde{\theta}\| w_\beta(\theta) \mathrm{d}\rho \\
&\leq \frac{\exp(-\beta(\inf_{B^c_{\theta,\tilde{r}}(\tilde{\theta})} J(\theta) - J_r))}{\rho(B_{\theta,r}(\tilde{\theta}))} \int_{B^c_{\theta,\tilde{r}}(\tilde{\theta})} \|\theta - \tilde{\theta}\| \mathrm{d}\rho \\
&\leq \frac{\exp(-\beta(\inf_{B^c_{\theta,\tilde{r}}(\tilde{\theta})} J(\theta) - J_r))}{\rho(B_{\theta,r}(\tilde{\theta}))} \int \|\theta - \tilde{\theta}\| \mathrm{d}\rho.
\end{aligned}
$$

We also notice

$$\inf_{B^c_{\theta,\tilde{r}}(\tilde{\theta})} J(\theta) - J_r \geq \min\{\delta_J + \underline{J}, (\eta\tilde{r})^{1/\mu} + \underline{J}\} - J_r \geq (\eta\tilde{r})^{1/\mu} - J_r + \underline{J} = q,$$

where the first inequality comes from Assumption 4.1 and the second inequality comes from the definition of $\tilde{r}$ and $q$, $\tilde{r} = \frac{(q+J_r-\underline{J})^\mu}{\eta} \leq \frac{\delta_J}{\eta}$. Combining the above inequality and the definition of $\tilde{r}$, we have

$$\|\mathcal{M}_\beta[\mu] - \tilde{\theta}\| \leq \frac{(q+J_r-\underline{J})^\mu}{\eta} + \frac{\exp(-\alpha(\inf_{B^c_{\theta,\tilde{r}}(\tilde{\theta})} J(\theta) - J_r))}{\rho(B_{\theta,r}(\tilde{\theta}))} \int \|\theta - \tilde{\theta}\| \mathrm{d}\rho$$

$$\leq \frac{(q+J_r-\underline{J})^\mu}{\eta} + \frac{\exp(-\beta q)}{\rho(B_{\theta,r}(\tilde{\theta}))} \int \|\theta - \tilde{\theta}\| \mathrm{d}\rho.$$

$\square$

Then we will establish a lower bound for $\rho_t(B_{\theta,r}(\tilde{\theta}))$. Notice that $\rho_t(B_{\theta,r}(\tilde{\theta})) = \rho_t\left(\{(\theta,\omega) : \|\theta - \tilde{\theta}\|^2 \leq r^2\}\right) \geq \rho_t\left(\{(\theta,\omega) : \|\theta - \tilde{\theta}\|^2 + m^{-1}\|\omega\|^2 \leq r^2\}\right) := \rho_t(B_r(\tilde{\theta}, 0))$. We first define the mollifier $\phi_r(\theta,\omega)$ as follows

$$\phi_r(\theta,\omega) = \begin{cases} \exp\left(1 - \dfrac{r^2}{r^2 - (\|\theta - \tilde{\theta}\|^2 + m^{-1}\|\omega\|^2)}\right), & \text{if } \|\theta - \tilde{\theta}\|^2 + m^{-1}\|\omega\|^2 \leq r^2, \\ 0, & \text{else.} \end{cases}$$

We have $\mathrm{Im}(\phi_r) \in [0, 1]$, $\phi_r \in \mathcal{C}_c^\infty$. First, we compute the first-order and second-order derivatives as

$$\nabla_\theta \phi_r = -2r^2 \frac{\theta - \tilde{\theta}}{\left(r^2 - (\|\theta - \tilde{\theta}\|^2 + m^{-1}\|\omega\|^2)\right)^2} \phi_r,$$

$$\nabla_\omega \phi_r = -2r^2 \frac{m^{-1}\omega}{\left(r^2 - (\|\theta - \tilde{\theta}\|^2 + m^{-1}\|\omega\|^2)\right)^2} \phi_r,$$

and

$$\Delta_\theta \phi_r = -2r^2 \frac{D\left(r^2 - (\|\theta - \tilde{\theta}\|^2 + m^{-1}\|\omega\|^2)\right)^2 - 2\left(r^2 - (\|\theta - \tilde{\theta}\|^2 + m^{-1}\|\omega\|^2)\right)\left(-2(\theta - \tilde{\theta}) \cdot (\theta - \tilde{\theta})\right)}{\left(r^2 - (\|\theta - \tilde{\theta}\|^2 + m^{-1}\|\omega\|^2)\right)^4} \phi_r$$

$$+ 4r^4 \frac{\|\theta - \tilde{\theta}\|^2}{\left(r^2 - (\|\theta - \tilde{\theta}\|^2 + m^{-1}\|\omega\|^2)\right)^4} \phi_r$$

$$= 2r^2 \frac{\|\theta - \tilde{\theta}\|^2(-2r^2 + 4\|\theta - \tilde{\theta}\|^2 + 4m^{-1}\|\omega\|^2) - D\left(r^2 - (\|\theta - \tilde{\theta}\|^2 + m^{-1}\|\omega\|^2)\right)^2}{\left(r^2 - (\|\theta - \tilde{\theta}\|^2 + m^{-1}\|\omega\|^2)\right)^4} \phi_r,$$

$$\Delta_\omega \phi_r = -2r^2 \frac{D\left(r^2 - (\|\theta - \tilde{\theta}\|^2 + m^{-1}\|\omega\|^2)\right)^2 m^{-1} - 2\left(r^2 - (\|\theta - \tilde{\theta}\|^2 + m^{-1}\|\omega\|^2)\right)(-m^{-1}\omega) \cdot (m^{-1}\omega)}{\left(r^2 - (\|\theta - \tilde{\theta}\|^2 + m^{-1}\|\omega\|^2)\right)^4} \phi_r$$

$$+ 4r^4 \frac{m^{-2}\omega^2}{\left(r^2 - (\|\theta - \tilde{\theta}\|^2 + m^{-1}\|\omega\|^2)\right)^4} \phi_r$$

$$= 2r^2 \frac{m^{-2}\|\omega\|^2(-2r^2 + 4\|\theta - \tilde{\theta}\|^2 + 4m^{-1}\|\omega\|^2) - D\left(r^2 - (\|\theta - \tilde{\theta}\|^2 + m^{-1}\|\omega\|^2)\right)^2 m^{-1}}{\left(r^2 - (\|\theta - \tilde{\theta}\|^2 + m^{-1}\|\omega\|^2)\right)^4} \phi_r.$$

**Lemma D.3.** *Let $T > 0, r > 0$. Choose parameters $\overline{\sigma} \geq \sigma(t) \geq \underline{\sigma} > 0$. Assume $\rho \in \mathcal{C}([0, T], \mathcal{P}(\mathbb{R}^{2D}))$ weakly solves the Fokker-Planck equation* (3) *with initial condition $\rho_0$. Then, $\forall t \in [0, T]$, we have*

$$\rho_t(B_{\theta,r}(\tilde{\theta})) \geq \left( \int \phi_r(\theta, \omega) \mathrm{d}\rho_0(\theta, \omega) \right) \exp(-pt),$$

*where*

$$p := \max \left\{ \frac{4\lambda(\sqrt{k}r + B)\sqrt{k}}{(1-k)^2 r} + \frac{2\overline{\sigma}^2(k+D)}{(1-k)^4 r^2}, \frac{8(B+r)^2 \lambda^2}{(2k-1)\underline{\sigma}^2} \right\},$$

*for any $B > 0$ with $\sup_{t \in [0,T]} \|\mathcal{M}_\beta[\mu_t] - \tilde{\theta}\| \leq B$ and for any $k \in (\frac{1}{2}, 1)$ satisfying $(-1 + 2k)k \geq 2D(1-k)^2$ and $\lambda = \max\{\gamma_1, m\}$.*

*Proof.* By the properties of the mollifier $\phi_r$, we have $\mu_t(B_{\theta,r}(\tilde{\theta})) \geq \rho_t(B_r(\tilde{\theta}, 0)) \geq \int \phi_r(\theta, \omega) \mathrm{d}\rho_t(\theta, \omega)$. Using properties of the weak solution $\rho_t$, we have

$$\frac{\mathrm{d}}{\mathrm{d}t} \int \phi_r(\theta, \omega) \mathrm{d}\rho_t(\theta, \omega) = \int \left\langle \omega - \gamma_1(\theta - \mathcal{M}_\beta[\mu_t]), -2r^2 \frac{\theta - \tilde{\theta}}{\left( r^2 - (\|\theta - \tilde{\theta}\|^2 + \|\omega\|^2) \right)^2} \phi_r \right\rangle \mathrm{d}\rho_t$$

$$+ \int \left\langle -m(\theta - \mathcal{M}_\beta[\mu_t]) - \gamma_2 \omega, -2r^2 \frac{m^{-1}\omega}{\left( r^2 - (\|\theta - \tilde{\theta}\|^2 + \|\omega\|^2) \right)^2} \phi_r \right\rangle \mathrm{d}\rho_t$$

$$+ \frac{\sigma^2}{2} \int (m\Delta_\omega \phi_r + \Delta_\theta \phi_r) \, \mathrm{d}\rho_t$$

$$= \int \left( \gamma\langle \theta - \mathcal{M}_\beta[\mu_t], \theta - \tilde{\theta} \rangle + \langle \gamma_2 \omega, m^{-1}\omega \rangle + \langle \omega, \tilde{\theta} - \mathcal{M}_\beta[\mu_t] \rangle \right)$$

$$\frac{2\,r^2}{\left( r^2 - (\|\theta - \tilde{\theta}\|^2 + m^{-1}\|\omega\|^2) \right)^2} \phi_r \mathrm{d}\rho_t + \frac{\sigma^2}{2} \int (m\Delta_\omega \phi_r + \Delta_\theta \phi_r) \, \mathrm{d}\rho_t$$

$$:= \int T_1(\theta, \omega) \mathrm{d}\rho_t + \int T_2(\theta, \omega) \mathrm{d}\rho_t.$$

Since $\phi_r$ vanishes outside of $D_r := \{(\theta, \omega) : \|\theta - \tilde{\theta}\|^2 + m^{-1}\|\omega\|^2 \leq r^2\}$, we restrict our attention to the open ball $D_r$. To obtain the lower bound, we introduce the following subsets

$$K_1 := \left\{ (\theta, \omega) : \|\theta - \tilde{\theta}\|^2 + m^{-1}\|\omega\|^2 > kr^2 \right\},$$

$$K_2 := \left\{ (\theta, \omega) : -\left( \gamma_1\langle \theta - \mathcal{M}_\beta[\mu_t], \theta - \tilde{\theta} \rangle + \langle \gamma_2 \omega, m^{-1}\omega \rangle + \langle \omega, \tilde{\theta} - \mathcal{M}_\beta[\rho_t] \rangle \right) (r^2 - \|\theta - \tilde{\theta}\| - m^{-1}\|\omega\|^2)^2 \right.$$

$$\left. > \tilde{k} \frac{\sigma^2}{2} r^2 (\|\theta - \tilde{\theta}\|^2 + m^{-1}\|\omega\|^2) \right\}, \tag{17}$$

where $\tilde{k} = 2k - 1 \in (0, 1)$. We divide the integral region into three domains.

**Domain** $\Omega_r \cap K_1^c$: We have $\|\theta - \tilde{\theta}\|^2 + m^{-1}\|\omega\|^2 \leq kr^2$ in this domain and we can get

$$
\begin{aligned}
T_1(\theta, \omega) &= \left( \langle \gamma_1(\theta - \mathcal{M}_\beta[\mu_t]), \theta - \tilde{\theta} \rangle + \langle \gamma_2\omega, m^{-1}\omega \rangle + \langle \omega, \tilde{\theta} - \mathcal{M}_\beta[\mu_t] \rangle \right) \frac{2r^2}{\left( r^2 - (\|\theta - \tilde{\theta}\|^2 + m^{-1}\|\omega\|^2) \right)^2} \phi_r \\
&\geq - \left( \gamma_1 \|\theta - \mathcal{M}_\beta[\mu_t]\| \|\theta - \tilde{\theta}\| + \|\omega\| \|\tilde{\theta} - \mathcal{M}_\beta[\mu_t]\| \right) \frac{2r^2}{\left( r^2 - (\|\theta - \tilde{\theta}\|^2 + m^{-1}\|\omega\|^2) \right)^2} \phi_r \\
&\geq - \left( \gamma_1 \|\theta - \tilde{\theta}\| + \|\omega\| \right) \frac{2(\sqrt{k}r + B)r^2}{\left( r^2 - (\|\theta - \tilde{\theta}\|^2 + m^{-1}\|\omega\|^2) \right)^2} \phi_r \\
&\geq - \left( \gamma_1 \|\theta - \tilde{\theta}\| + \|\omega\| \right) \frac{2(\sqrt{k}r + B)r^2}{(r^2 - kr^2)^2} \phi_r \\
&\geq - \lambda \left( \|\theta - \tilde{\theta}\| + m^{-1}\|\omega\| \right) \frac{2(\sqrt{k}r + B)}{(1-k)^2 r^2} \phi_r \\
&\geq - 2\lambda\sqrt{k}r \frac{2(\sqrt{k}r + B)}{(1-k)^2 r^2} \phi_r = -\frac{4\lambda(\sqrt{k}r + B)\sqrt{k}}{(1-k)^2 r} \phi_r := -p_1\phi_r,
\end{aligned}
$$

where the first inequality comes from Cauchy-Schwarz inequality and the positiveness of $\|\omega\|^2$, the second inequality comes from the boundedness of $\|\theta - \mathcal{M}_\beta[\mu_t]\| \leq \|\theta - \tilde{\theta}\| + \|\tilde{\theta} - \mathcal{M}_\beta[\mu_t]\| \leq \sqrt{k}r + B$, the third inequality comes from the defintion of domain $K_1^c$, and the fourth inequality comes from the definition of $\lambda$.

For $T_2$, we have

$$
\begin{aligned}
T_2(\theta, \omega) =& \sigma^2 r^2 \frac{(\|\theta - \tilde{\theta}\|^2 + m^{-1}\|\omega\|^2)(-2r^2 + 4\|\theta - \tilde{\theta}\|^2 + 4m^{-1}\|\omega\|^2)}{\left( r^2 - (\|\theta - \tilde{\theta}\|^2 + m^{-1}\|\omega\|^2) \right)^4} \phi_r \\
&- \sigma^2 r^2 \frac{2D}{\left( r^2 - (\|\theta - \tilde{\theta}\|^2 + m^{-1}\|\omega\|^2) \right)^2} \phi_r \\
\geq& - \sigma^2 r^2 \frac{2r^2(\|\theta - \tilde{\theta}\|^2 + m^{-1}\|\omega\|^2)}{\left( r^2 - (\|\theta - \tilde{\theta}\|^2 + m^{-1}\|\omega\|^2) \right)^4} \phi_r \\
&- \sigma^2 r^2 \frac{2D}{\left( r^2 - (\|\theta - \tilde{\theta}\|^2 + m^{-1}\|\omega\|^2) \right)^2} \phi_r \\
\geq& - \sigma^2 r^2 \frac{2kr^4}{(r^2 - kr^2)^4} \phi_r - \sigma^2 r^2 \frac{2D}{(r^2 - kr^2)^2} \phi_r \\
=& - \sigma^2 \frac{2k}{(1-k)^4 r^2} \phi_r - \sigma^2 \frac{2D}{(1-k)^2 r^2} \phi_r \\
\geq& - \frac{2\overline{\sigma}^2(k + D)}{(1-k)^4 r^2} \phi_r := -p_2\phi_r,
\end{aligned}
$$

(18)

where the first inequlaity uses the positiveness of $\|\theta - \tilde{\theta}\|^2$ and $\|\omega\|^2$, the second inequality uses the properties of $K_1^c$, and the third inequality use $1 - k \in (0, \frac{1}{2})$.

**Domain** $\Omega_r \cap K_1 \cap K_2^c$

In this domian, we have $\|\theta - \tilde{\theta}\|^2 + m^{-1}\|\omega\|^2 > kr^2$ and

$$
- \left( \gamma_1 \langle \theta - \mathcal{M}_\beta[\rho_t], \theta - \tilde{\theta} \rangle + \langle \gamma_2\omega, m^{-1}\omega \rangle + \langle \omega, \tilde{\theta} - \mathcal{M}_\beta[\rho_t] \rangle \right)(r^2 - \|\theta - \tilde{\theta}\| - m^{-1}\|\omega\|^2)^2 \leq \tilde{k} \frac{\sigma^2}{2} r^2 (\|\theta - \tilde{\theta}\|^2 + m^{-1}\|\omega\|^2).
$$

(19)

Our goal is to show $T_1(\theta, \omega) + T_2(\theta, \omega) \geq 0$ in this subset. We first compute

$$\frac{T_1(\theta, \omega) + T_2(\theta, \omega)}{2r^2 \phi_r} \left( r^2 - (\|\theta - \tilde{\theta}\|^2 + m^{-1}\|\omega\|^2) \right)^4$$

$$= \left( \langle \gamma(\theta - \mathcal{M}_\beta[\mu_t]), \theta - \tilde{\theta} \rangle + \langle \gamma_2 \omega, m^{-1}\omega \rangle + \langle \omega, \tilde{\theta} - \mathcal{M}_\beta[\mu_t] \rangle \right) \left( r^2 - (\|\theta - \tilde{\theta}\|^2 + m^{-1}\|\omega\|^2) \right)^2$$

$$+ \sigma^2 m^{-1}\|\omega\|^2 (-r^2 + 2\|\theta - \tilde{\theta}\|^2 + 2m^{-1}\|\omega\|^2)$$

$$- \frac{\sigma^2 D}{2} \left( r^2 - (\|\theta - \tilde{\theta}\|^2 + m^{-1}\|\omega\|^2) \right)^2$$

$$+ \sigma^2 \|\theta - \tilde{\theta}\|^2 (-r^2 + 2\|\theta - \tilde{\theta}\|^2 + 2m^{-1}\|\omega\|^2)$$

$$- \frac{\sigma^2 D}{2} \left( r^2 - (\|\theta - \tilde{\theta}\|^2 + m^{-1}\|\omega\|^2) \right)^2.$$

To prove the positiveness, we need to prove

$$- \left( \langle \gamma(\theta - \mathcal{M}_\beta[\mu_t]), \theta - \tilde{\theta} \rangle + \langle \gamma_2 \omega, m^{-1}\omega \rangle + \langle \omega, \tilde{\theta} - \mathcal{M}_\beta[\mu_t] \rangle \right) \left( r^2 - (\|\theta - \tilde{\theta}\|^2 + m^{-1}\|\omega\|^2) \right)^2$$

$$+ \sigma^2 D \left( r^2 - (\|\theta - \tilde{\theta}\|^2 + m^{-1}\|\omega\|^2) \right)^2$$

$$\leq \sigma^2 (m^{-1}\|\omega\|^2 + \|\theta - \tilde{\theta}\|^2)(-r^2 + 2\|\theta - \tilde{\theta}\|^2 + 2m^{-1}\|\omega\|^2).$$

For the first term, we have

$$- \left( \langle \gamma(\theta - \mathcal{M}_\beta[\mu_t]), \theta - \tilde{\theta} \rangle + \langle \gamma_2 \omega, m^{-1}\omega \rangle + \langle \omega, \tilde{\theta} - \mathcal{M}_\beta[\mu_t] \rangle \right) \left( r^2 - (\|\theta - \tilde{\theta}\|^2 + m^{-1}\|\omega\|^2) \right)^2$$

$$\leq \tilde{k} \frac{\sigma^2}{2} r^2 (\|\theta - \tilde{\theta}\|^2 + m^{-1}\|\omega\|^2])$$

$$= (2k - 1)\frac{\sigma^2}{2} r^2 (\|\theta - \tilde{\theta}\|^2 + m^{-1}\|\omega\|^2)$$

$$\leq (2\|\theta - \tilde{\theta}\|^2 + 2m^{-1}\|\omega\|^2 - r^2)\frac{\sigma^2}{2}(\|\theta - \tilde{\theta}\|^2 + m^{-1}\|\omega\|^2),$$

where the first inequality comes from the positiveness of the norm and the second inequality comes from Equation (19). By the definition $\tilde{k} = 2k - 1$, we have the equality in the fourth line and the last inequality comes from $kr^2 \leq \|\theta - \tilde{\theta}\|^2 + m^{-1}\|\omega\|^2$. For the second term, we have

$$\sigma^2 D \left( r^2 - (\|\theta - \tilde{\theta}\|^2 + m^{-1}\|\omega\|^2) \right)^2$$

$$\leq \sigma^2 D (1 - k)^2 r^4$$

$$\leq \frac{\sigma^2}{2}(2k - 1)r^2 k r^2$$

$$\leq \frac{\sigma^2}{2}(2\|\theta - \tilde{\theta}\|^2 + 2m^{-1}\|\omega\|^2 - r^2)(\|\theta - \tilde{\theta}\|^2 + m^{-1}\|\omega\|^2),$$

where in the second inequality we use $(-1 + 2k)k \geq 2D(1 - k)^2$. Hence, we have the positiveness of $T_1(\theta, \omega) + T_2(\theta, \omega)$.

**Domain** $\Omega_r \cap K_1 \cap K_2$

In this subset, we have $\|\theta - \tilde{\theta}\|^2 + m^{-1}\|\omega\|^2 > kr^2$ and

$$- \left( \gamma_1 \langle \theta - \mathcal{M}_\beta[\mu_t], \theta - \tilde{\theta} \rangle + \langle \gamma_2 \omega, m^{-1}\omega \rangle + \langle \omega, \tilde{\theta} - \mathcal{M}_\beta[\rho_t] \rangle \right) (r^2 - \|\theta - \tilde{\theta}\| - m^{-1}\|\omega\|^2)^2$$

$$> \tilde{k}\frac{\sigma^2}{2} r^2 (\|\theta - \tilde{\theta}\|^2 + m^{-1}\|\omega\|^2). \tag{20}$$

In this subset, we have

$$
\begin{aligned}
&\gamma_1 \langle \theta - \mathcal{M}_\beta[\mu_t], \theta - \tilde{\theta} \rangle + \langle \gamma_2 \omega, m^{-1}\omega \rangle + \langle \omega, \tilde{\theta} - \mathcal{M}_\beta[\mu_t] \rangle \\
&\geq \gamma_1 \langle \theta - \mathcal{M}_\beta[\mu_t], \theta - \tilde{\theta} \rangle + \langle \omega, \tilde{\theta} - \mathcal{M}_\beta[\mu_t] \rangle \\
&\geq -\gamma_1 \| \theta - \mathcal{M}_\beta[\mu_t] \| \| \theta - \tilde{\theta} \| - \| \omega \| \| \tilde{\theta} - \mathcal{M}_\beta[\mu_t] \| \\
&\geq -\lambda (\| \tilde{\theta} - \mathcal{M}_\beta[\mu_t] \| + \| \theta - \tilde{\theta} \|)(\| \theta - \tilde{\theta} \| + m^{-1}\| \omega \|),
\end{aligned}
\tag{21}
$$

where the last inequality comes from the definition of $\lambda$. From the inequality (20), we have

$$
\begin{aligned}
\frac{(\| \theta - \tilde{\theta} \| + m^{-1}\| \omega \|)^2}{(r^2 - \| \theta - \tilde{\theta} \|^2 - m^{-1}\| \omega \|^2)^2} &\leq 2 \frac{\| \theta - \tilde{\theta} \|^2 + m^{-1}\| \omega \|^2}{(r^2 - \| \theta - \tilde{\theta} \|^2 - m^{-1}\| \omega \|^2)^2} \\
&< -\frac{4}{\tilde{k}\sigma^2 r^2} \left( \gamma_1 \langle \theta - \mathcal{M}_\beta[\rho_t], \theta - \tilde{\theta} \rangle + \langle \gamma_2 \omega, m^{-1}\omega \rangle + \langle \omega, \tilde{\theta} - \mathcal{M}_\beta[\rho_t] \rangle \right).
\end{aligned}
\tag{22}
$$

Then we are ready to prove

$$
\begin{aligned}
&\frac{\langle \gamma_1(\theta - \mathcal{M}_\beta[\mu]), \theta - \tilde{\theta} \rangle + \langle \gamma_2 \omega, m^{-1}\omega \rangle + \langle \omega, \tilde{\theta} - \mathcal{M}_\beta[\mu_t] \rangle}{\left( r^2 - (\| \theta - \tilde{\theta} \|^2 + m^{-1}\| \omega \|^2) \right)^2} \\
&\geq -\lambda \frac{(\| \tilde{\theta} - \mathcal{M}_\beta[\mu_t] \| + \| \tilde{\theta} - \theta \|)(\| \theta - \tilde{\theta} \| + m^{-1}\| \omega \|)}{\left( r^2 - (\| \theta - \tilde{\theta} \|^2 + m^{-1}\| \omega \|^2) \right)^2} \\
&\geq \lambda \frac{4}{\tilde{k}\sigma^2 r^2} \frac{\left( \gamma_1 \langle \theta - \mathcal{M}_\beta[\rho_t], \theta - \tilde{\theta} \rangle + \langle \gamma_2 \omega, m^{-1}\omega \rangle + \langle \omega, \tilde{\theta} - \mathcal{M}_\beta[\rho_t] \rangle \right)(\| \tilde{\theta} - \mathcal{M}_\beta[\mu_t] \| + \| \tilde{\theta} - \theta \|)}{\| \theta - \tilde{\theta} \| + m^{-1}\| \omega \|} \\
&\geq -\lambda^2 \frac{4}{\tilde{k}\sigma^2 r^2} \frac{(\| \tilde{\theta} - \mathcal{M}_\beta[\mu_t] \| + \| \tilde{\theta} - \theta \|)^2 \left( \| \theta - \tilde{\theta} \| + m^{-1}\| \omega \| \right)}{\| \theta - \tilde{\theta} \| + m^{-1}\| \omega \|} \\
&\geq -\lambda^2 \frac{4}{\tilde{k}\sigma^2 r^2} (\| \tilde{\theta} - \mathcal{M}_\beta[\mu_t] \| + \| \tilde{\theta} - \theta \|)^2 \\
&\geq -\lambda^2 \frac{4(B+r)^2}{\tilde{k}\sigma^2 r^2} a,
\end{aligned}
$$

where the first and third inequalities are derived from the inequality (21), and the second one is a consequence of (22). Utilizing Cauchy–Schwarz inequality and the definition specified in $\lambda$, we have the third and fourth inequalities.

Given this we have

$$
\begin{aligned}
T_1(\theta, \omega) &= \frac{\langle \gamma_1 \theta - \mathcal{M}_\beta[\mu_t], \theta - \tilde{\theta} \rangle + \langle \gamma_2 \omega, m^{-1}\omega \rangle + \langle \omega, \tilde{\theta} - \mathcal{M}_\beta[\mu_t] \rangle}{\left( r^2 - (\| \theta - \tilde{\theta} \|^2 + m^{-1}\| \omega \|^2) \right)^2} 2r^2 \phi_r \\
&\geq -\frac{8(B+r)^2 \lambda^2}{\tilde{k}\underline{\sigma}^2} \phi_r = -\frac{8(B+r)^2 \lambda^2}{(2k-1)\underline{\sigma}^2} \phi_r := -p_3 \phi_r.
\end{aligned}
$$

For $T_2$, it is positive whenever

$$
(\| \theta - \tilde{\theta} \|^2 + m^{-1}\| \omega \|^2)(-2r^2 + 4\| \theta - \tilde{\theta} \| + 4m^{-1}\| \omega \|^2) \geq 2D \left( r^2 - (\| \theta - \tilde{\theta} \|^2 + m^{-1}\| \omega \|^2) \right)^2,
$$

we have

$$
(\|\theta - \tilde{\theta}\|^2 + m^{-1}\|\omega\|^2)(-2r^2 + 4\|\theta - \tilde{\theta}\| + 4m^{-1}\|\omega\|^2)
$$
$$
\geq (\|\theta - \tilde{\theta}\|^2 + m^{-1}\|\omega\|^2)(-2r^2 + 4kr^2)
$$
$$
\geq (\|\theta - \tilde{\theta}\|^2 + m^{-1}\|\omega\|^2)(-1 + 2k)2r^2
$$
$$
\geq (\|\theta - \tilde{\theta}\|^2 + m^{-1}\|\omega\|^2)2D\frac{(1-k)^2}{k}2r^2
$$
$$
\geq r^2 2D(1-k)^2 2r^2
$$
$$
= 2D(r^2 - kr^2)^2
$$
$$
\geq 2D(r^2 - (\|\theta - \tilde{\theta}\|^2 + m^{-1}\|\omega\|^2))^2.
$$

This is safiesifed for all $\|\theta - \tilde{\theta}\|^2 + m^{-1}\|\omega\|^2 \geq kr^2$.

**Concluding the proof:** Using the evolution of $\phi_r$, we now get

$$
\frac{\mathrm{d}}{\mathrm{dt}}\int \phi_r(\theta,\omega)\mathrm{d}\rho_t(\theta,\omega) = \int_{K_1 \cap K_2^c \cap \Omega_r} T_1(\theta,\omega) + T_2(\theta,\omega)\mathrm{d}\rho_t(\theta,\omega)
$$
$$
+ \int_{K_1 \cap K_2 \cap \Omega_r} T_1(\theta,\omega) + T_2(\theta,\omega)\mathrm{d}\rho_t(\theta,\omega) + \int_{K_1^c \cap \Omega_r} T_1(\theta,\omega) + T_2(\theta,\omega)\mathrm{d}\rho_t(\theta,\omega)
$$
$$
\geq -\max\{p_1 + p_2, p_3\}\int \phi_r(\theta,\omega)\mathrm{d}\rho_t(\theta,\omega) = -p\int \phi_r(\theta,\omega)\mathrm{d}\rho_t(\theta,\omega).
$$

$\square$

*Proof of Theorem 4.5.* We choose parameters $\beta$ such that

$$
\beta > \beta_0 := \frac{1}{q_\epsilon}\left(\log\left(\frac{4\sqrt{2E[\rho_0]}}{\mathrm{c}(\tau,\lambda)\sqrt{\epsilon}} + \frac{p}{(1-\tau)\lambda}\log\left(\frac{E[\rho_0]}{\epsilon}\right) - \log\rho_0(B_{\frac{r_\epsilon}{2}}(\tilde{\theta},0))\right)\right),
$$

where we introduce

$$
\mathrm{c}(\tau,\lambda) = \frac{\tau\gamma}{\lambda}, \quad q_\epsilon = \frac{1}{2}\min\left\{\left(\frac{\mathrm{c}(\tau,\lambda)\sqrt{\epsilon}\eta}{2}\right)^{1/\mu}, \delta_J\right\}, \quad \text{and } r_\epsilon = \max_{x\in[0,R_0]}\{\max_{(\theta,\omega)\in B_s(\tilde{\theta},0)} J(\theta) \leq q_\epsilon + \underline{J}\},
$$

and define the time horizon $T_\beta \geq 0$, which may depend on $\beta$, by

$$
T_\beta = \sup\{t \geq 0 : E[\mu_{t'}] > \epsilon \text{ and } \|\mathcal{M}_\beta[\mu_{t'}] - \tilde{\theta}\| < C(t') \text{ for all } t' \in [0,t]\}
$$

with $C(t) = \mathrm{c}(\tau,\lambda)\sqrt{E(\rho_t)}$. First we want to prove $T_\beta > 0$, which follows from the continuity of the mappings $t \to E[\rho_t]$ and $t \to \|\mathcal{M}_\beta[\mu_t] - \tilde{\theta}\|$ since $E[\rho_0] > 0$ and $\|\mathcal{M}_\beta[\mu_0] - \tilde{\theta}\| < C(0)$. While the former holds by assumption, the latter follows by

$$
\|\mathcal{M}_\beta[\mu_0] - \tilde{\theta}\| \leq \frac{(q_\epsilon + J_{r_\epsilon} - \underline{J})^\mu}{\eta} + \frac{\exp(-\beta q_\epsilon)}{\rho(B_{\theta,r_\epsilon}(\tilde{\theta}))}\int \|\theta - \tilde{\theta}\|\mathrm{d}\rho_0(\theta,\omega)
$$
$$
\leq \frac{(q_\epsilon + J_{r_\epsilon} - \underline{J})^\mu}{\eta} + \frac{\exp(-\beta q_\epsilon)}{\rho(B_{r_\epsilon}(\tilde{\theta},0))}\int \|\theta - \tilde{\theta}\|\mathrm{d}\rho_0(\theta,\omega)
$$
$$
\leq \frac{\mathrm{c}(\tau,\lambda)\sqrt{\epsilon}}{2} + \frac{\exp(-\beta q_\epsilon)}{\rho(B_{r_\epsilon}(\tilde{\theta},0))}\sqrt{2E[\rho_0]}
$$
$$
\leq \mathrm{c}(\tau,\lambda)\sqrt{\epsilon} \leq \mathrm{c}(\tau,\lambda)\sqrt{E[\rho_0]} = C(0),
$$

where we use the definition of $\beta$ in the first inequality of the last line. Recall the Lemma D.1, up to time $T_\beta$

$$
\frac{\mathrm{d}}{\mathrm{dt}}E[\rho_t] \leq -\gamma E[\rho_t] + \lambda\sqrt{E[\rho_t]}\|\mathcal{M}_\beta[\mu_t] - \tilde{\theta}\| + \frac{\sigma^2(t)D(m+1)}{2}
$$
$$
\leq -(1-\tau)\gamma E[\rho_t] + \frac{\sigma^2(t)D(m+1)}{2}.
$$

Thus we have

$$\frac{\mathrm{d}}{\mathrm{d}t}\left(\exp\left((1-\tau)\gamma t\right)E[\rho_t]\right) = (1-\tau)\gamma\left(\exp\left((1-\tau)\gamma t\right)E[\rho_t]\right) + \exp\left((1-\tau)\gamma t\right)\frac{\mathrm{d}}{\mathrm{d}t}E[\rho_t]$$

$$\leq \exp\left((1-\tau)\gamma t\right)\frac{\sigma^2(t)D(m+1)}{2}.$$

Therefore we have

$$\left(\exp\left((1-\tau)\gamma t\right)E[\rho_t]\right) - E[\rho_0] \leq \int_0^t \exp\left((1-\tau)\lambda s\right)\sigma^2(s)\mathrm{d}s$$

$$= \frac{\sigma_1^2(1-\exp\left((-2\sigma_2 + \lambda(1-\tau))t\right))}{2\sigma_2 - \lambda(1-\tau)}.$$

We can get the boundedness for $E[\rho_t]$, for $2\sigma_2 - \lambda(1-\tau) < 0$ by the chosen of $\tau$ and $\lambda$, then we have

$$E[\rho_t] \leq \exp\left(-(1-\tau)t\lambda\right)E[\rho_0].$$

Accordingly, we note that $E(\rho_t)$ is decreasing in $t$, which implies the decay of the function $C(t)$ as well. Hence, recalling the definition of $T_\beta$, we may bound $\max_{t\in[0,T_\beta]}\|\mathcal{M}_\beta[\rho_{t'}] - \tilde{\theta}\| \leq \max_{t\in[0,T_\beta]}C(t) \leq C(0)$. We now conclude by showing $\min_{t\in[0,T_\beta]}E(\rho_t) \leq \epsilon$ with $T_\beta \leq T^*$. For this, we distinguish the following three cases.

**Case $T_\beta \geq T^*$**: If $T_\beta \geq T^*$, we can use the definition of $T^* = \frac{1}{(1-\tau)\lambda}\log(\frac{E[\rho_0]}{\epsilon})$ and the time evolution bound of $E[\rho_t]$ to conclude that $E[\rho_{T^*}] \leq \epsilon$. Hence, by definition of $T_\beta$, we find $E[\rho_{T_\beta}] \leq \epsilon$ and $T_\beta = T^*$.

**Case $T_\beta < T^*$ and $E[\rho_{T_\beta}] \leq \epsilon$**: Nothing need to discussed in this case.

**Case $T_\beta < T^*$ and $E[\rho_{T_\beta}] > \epsilon$**: We shall prove that this case will never occur.

$$\|\mathcal{M}_\beta[\mu_{T_\beta}] - \tilde{\theta}\| \leq \frac{(q_\epsilon + J_{r_\epsilon} - \underline{J})^\mu}{\eta} + \frac{\exp(-\beta q_\epsilon)}{\rho(B_{\theta,r_\epsilon}(\tilde{\theta}))}\int\|\theta - \tilde{\theta}\|\mathrm{d}\rho_{T_\beta}(\theta,\omega)$$

$$< \frac{c(\tau,\lambda)\sqrt{E[\rho_{T_\beta}]}}{2} + \frac{\exp(-\beta q_\epsilon)}{\rho(B_{\theta,r_\epsilon}(\tilde{\theta}))}\sqrt{E[\mu_{T_\beta}]}.$$

Since, we have $\max_{t\in[0,T_\beta]}\|\mathcal{M}_\beta[\mu_{t'}] - \tilde{\theta}\| = B = C(0)$ guarantees that there exist a $p > 0$ with

$$\rho_{T_\beta}(B_{\theta,r_\epsilon}(\tilde{\theta})) \geq \left(\int\phi_{r_\epsilon}(\theta,\omega)\mathrm{d}\rho_0(\theta,\omega)\right)\exp(-pT_\beta) \geq \frac{1}{2}\rho_0\left(B_{\frac{r_\epsilon}{2}}(\tilde{\theta},0)\right)\exp(-pT^*), \tag{23}$$

where we used $(\tilde{\theta}, 0) \in supp(\rho_0)$ for bounding the initial mass $\rho_0$ and the fact that $\phi_r$ is bounded from below on $B_{\frac{r_\epsilon}{2}}(\tilde{\theta}, 0)$ by $1/2$. With this, we can conclude that

$$\|\mathcal{M}_\beta[\mu_{T_\beta}] - \tilde{\theta}\| < \frac{c(\tau,\lambda)\sqrt{E[\rho_{T_\beta}]}}{2} + \frac{2\exp(-\beta q_\epsilon)}{\rho(B_{\frac{r_\epsilon}{2}}(\tilde{\theta},0))\exp(-pT^*)}\sqrt{E[\rho_{T_\beta}]}$$

$$\leq c(\tau,\lambda)\sqrt{E[\rho_{T_\beta}]} = C(T_\beta),$$

where the first inequality in the last line holds by the choice of $\beta$. This establishes the desired contradiction, against the consequence of the continuity of the mappings $t \to E[\rho_t]$ and $t \to \|\mathcal{M}_\beta[\mu_t] - \tilde{\theta}\|$. $\qquad\square$

# E. Simulation Details

## E.1. Linear-quadratic-Gaussian Control Problem

We begin by considering a classical LQG control problem, where the state dynamics is governed by:

$$\mathrm{d}\mathbf{x}_t = 2\boldsymbol{\alpha}_t\mathrm{d}t + \sqrt{2}\mathrm{d}W_t,$$

incorporating $t \in [0, T]$ and $\mathbf{x}_0 = x$. The cost functional is given by $J(\boldsymbol{\alpha}_t) = \mathbb{E}\left[\int_0^T\|\boldsymbol{\alpha}_r\|^2dt + g(X_T)\right]$. Here, the state process $\mathbf{x}_t$ is a $d$-dimensional vector, while the action process $\boldsymbol{\alpha}_t$ is a $d$-dimensional vector-valued function. The value

function $u$ can be defined as

$$u(t, \mathbf{x}) = \inf_{\boldsymbol{\alpha}} \mathbb{E}\left[\int_t^T f(t, \mathbf{x}_t, \boldsymbol{\alpha}_t, t)\mathrm{d}t + g(\mathbf{x}_T)\,|\mathbf{x}_t = \mathbf{x}\right].$$

By solving the Hamilton–Jacobi–Bellman equation for $u$, one can derive an explicit solution with the terminal condition $u(T, x) = g(x)$, given by

$$u(t, \mathbf{x}) = -\ln\left(\mathbb{E}\left[\exp\left(-g\left(\mathbf{x} + \sqrt{2}\mathbf{W}_{T-t}\right)\right)\right]\right).$$

## E.2. Ginzburg-Landau Model

In this model, superconducting electrons are described by a "macroscopic" wavefunction, $\varphi(z)$, with Landau free energy $\frac{\mu}{4}\int_0^1 |1 - \varphi(z)^2|^2\mathrm{d}z$. In order to add fluctuations (local variations in the wavefunction) to this model, Ginzburg suggested adding a term proportional to $|\nabla_z\varphi(z)|^2$, which can be interpolated as the kinetic energy term in quantum mechanics or the lowest order fluctuation term allowed by the symmetry of the order parameter. Adding this term to the free energy, we have the Ginzburg-Landau theory in zero field,

$$U[\varphi] = \frac{\lambda}{2}\int_0^1 |\nabla_z\varphi(z)|^2\mathrm{d}z + \frac{\mu}{4}\int_0^1 |1 - \varphi(z)^2|^2\mathrm{d}z.$$

Upon discretizing the space into $d + 1$ points, the potential is defined as

$$U(\varphi) = U(x_1, \cdots, x_d) := \left[\frac{\lambda}{2}\sum_{i=1}^{d+1}\left(\frac{x_i - x_{i-1}}{h}\right)^2 + \frac{\mu}{4}\sum_{i=1}^d (1 - x_i^2)^2\right]h,$$

where $x_i = \varphi(\frac{i}{d+1})$ for $i = 0, \cdots, d+1$ and $x_0 = x_{d+1} = 0$. The dynamics is given by

$$\mathrm{d}\mathbf{x}_t = \mathbf{b}(\mathbf{x}_t, \alpha_t)\mathrm{d}t + \sqrt{2}\mathrm{d}\mathbf{W}_t,$$

where the drift term is defined as

$$b(\mathbf{x}, a) = -\nabla_{\mathbf{x}}U(\mathbf{x}) + 2\alpha\boldsymbol{\omega}.$$

Here the potential is defined as

$$U(\varphi) = U(x_1, \cdots, x_d) := \left[\frac{\lambda}{2}\sum_{i=1}^{d+1}\left(\frac{x_i - x_{i-1}}{h}\right)^2 + \frac{\mu}{4}\sum_{i=1}^d (1 - x_i^2)^2\right]h,$$

and $\alpha$ is a scalar-valued function, represents the strength of the external field and the vector $\boldsymbol{\omega}$ is a $d$-dimensional vector represents the domian of the external field applied, the $i$-th element takes the value of 1 if the condition $\frac{i}{d+1} \in [0.25, 0.6]$ is satisfied, and 0 under other circumstances. The cost functional is defined

$$J[\alpha] = \mathbb{E}\left[\int_0^T \frac{1}{d}\|\mathbf{x}_t\|^2 + \|\alpha\|\mathrm{d}t + \frac{10}{d}\|\mathbf{x}_T\|^2\right].$$

## E.3. Systemic Risk Mean Field Control

We describe this problem as a network of $N$ banks, where $x_i$ denotes the logarithm of the cash reserves of the $i$-th bank. The following model introduces borrowing and lending between banks, given by:

$$\mathrm{d}\mathbf{x}_t = [\kappa(\bar{\mathbf{x}}_t - \mathbf{x}_t) + \boldsymbol{\alpha}_t]\mathrm{d}t + \sigma\mathrm{d}\mathbf{W}_t,$$

where $\bar{\mathbf{x}}_t = \frac{1}{n}\sum_{i=1}^n \mathbf{x}_t^i$ represents the average logarithm of the cash reserves across all banks. The control of the representative bank, i.e., the amount lent or borrowed at time $t$ is denoted by $\boldsymbol{\alpha}_t$. Based on the Almgren-Chriss linear price impact model, the running cost and terminal cost are given by:

$$f(x, \bar{x}, \alpha) = \frac{1}{2}\alpha^2 - q\alpha(\bar{x} - x) + \frac{\eta}{2}(\bar{x} - x)^2, g(x, \bar{x}) = \frac{c}{2}(\bar{x} - x)^2,$$

where $\eta$ and $c$ balance the individual bank's behavior with the average behavior of the other banks. $q$ weights the contribution of the components and helps to determine the sign of the control (i.e., whether to borrow or lend). Specifically, if the logarithmic cash reserve of an individual bank is smaller than the empirical mean, the bank will seek to borrow, choosing $\boldsymbol{\alpha}_t > 0$, and vice versa. We test the performance of our method with parameters $c = 2$, $k = 0.6$, and $\eta = 2$.

**Multi-Agent Robotic Systems**

To formulate the original path planning problem as a stochastic optimal control problem, we consider $n$ agents with states following the stochastic dynamics for $t \in [0, 1]$:

$$\mathrm{d}\mathbf{x}_t^i = 10\boldsymbol{\alpha}_t^i \mathrm{d}t + \mathrm{d}\mathbf{W}_t^i,$$

where represents the feedback control policy implemented as a neural network with parameters $\theta$, and are independent Wiener processes modeling noise. The environment contains $K$ circular obstacles centered at $\mathbf{y}^j$ with radius $r^j$. These are incorporated through the running cost:

$$f(t, \mathbf{x}, \boldsymbol{\alpha}) = \sum_{i=1}^n |\boldsymbol{\alpha}^i|^2 + \sum_{i=1}^n \sum_{j=1}^K s(|\mathbf{x}^i - \mathbf{y}^j|, r^j),$$

where $s(d, r)$-smooth penalty function defined piecewise as:

$$s(d, r) = \begin{cases} 1 & \text{if } d \leq r, \\ 0.5 + 0.5 \cos\left(\pi \frac{d-r}{0.2r}\right) & \text{if } r < d \leq 1.2r, \\ 0 & \text{otherwise.} \end{cases}$$

The terminal cost at final timeis given by the squared Euclidean distance between each agent's final position and its designated target.

## F. Neural Network Structure

In this subsection, we briefly illustrate the network structure used to model the action function. For the LQG and Ginzburg-Landau model problems, we employ traditional fully connected neural networks with a depth of 5 layers and a width of $5d$, where $d$ represents the dimension of the problem. In the mean-field control problem, we use the cylindrical type mean field neural network structure proposed in (Pham & Warin, 2024), where the control $\alpha(t, \mathbf{x})$ is parameterized as

$$\boldsymbol{\alpha}^i(t, \mathbf{x}; \theta) = \Psi\left[x^i, \frac{1}{n}\sum_{i=1}^n \psi(x_i; \theta_2); \theta_1\right],$$

where $\theta = \{\theta_1, \theta_2\}$. The advantage of this type of network is its extendibility, i.e, as demonstrated in the numerical results, control policies trained on a small number of agents $N$ can be effectively applied to problems with different values of $n$.

