# OpenReview forum: "Consensus Based Stochastic Optimal Control"
_ICML.cc/2025/Conference — ICML 2025 poster_

### Official Review · Reviewer_tREQ · 2025-02-15

**Overall Recommendation:** 3

**Summary:**

This paper proposes the Momentum Consensus-Based Optimization (M-CBO) and Adaptive Momentum Consensus-Based Optimization (Adam-CBO) method to solve the stochastic optimal control problem. While the numerical results are nice, I do not think the theoretical analysis is significant. I will put details in “Theoretical Claims”.

**Claims And Evidence:**

Yes

**Essential References Not Discussed:**

No

**Experimental Designs Or Analyses:**

I think the experiments are well designed.

**Methods And Evaluation Criteria:**

Yes

**Other Comments Or Suggestions:**

None

**Other Strengths And Weaknesses:**

The presentation of the paper is nice.

The weakness is already addressed before.

**Questions For Authors:**

Does the first equation in (1) lack a dt?
In sec 3.2, does the square refer to elementwise square (which result in a vector)? Please clarify.

**Relation To Broader Scientific Literature:**

The paper cites relevant literature.

**Theoretical Claims:**

I think the first item in Assumption 4.1.4 is too strong, excluding most of the optimal control problems of interest. The assumption $$J(\theta)-J(\tilde{\theta}) \ge \eta ||\theta-\tilde{\theta}||$$ implies that the landscape of $J(\theta)$ is a kink (very sharp) near $\tilde{\theta}$. Otherwise, if J is smooth, making a local Taylor expansion (note that $\nabla J(\tilde\theta) = 0$)
$$J(\theta) = J(\tilde\theta) + \nabla J(\tilde\theta)^\top (\theta - \tilde\theta) + o(\theta - \tilde\theta) = J(\tilde\theta) + o(\theta - \tilde\theta),$$
we get a result contradict to the assumption.

While there might be examples satisfying this assumption, I think most of the commonly studied control problems, such as LQR, do not satisfy this assumption. Therefore, the theoretical contribution is not that significant.

---

> ### Author Rebuttal · Authors · 2025-03-29
>
> Thank you for your thoughtful review and valuable feedback.
>
> 1. **On “ Assumption 4.1.4”**
>
>     Indeed, the assumption can be generalized as
>     $$\|\theta - \tilde\theta\| \leq \frac{(\mathcal{J} -\underline{ J})^\mu}{\eta}$$
>     for some $\mu,\eta>0$ .
>     When $\mu=\frac{1}{2}$, the condition is in general correct if $J$ is continuous near the minimum points, becase
>     $$
>     J(\theta) = J(\tilde \theta) + \nabla J(\tilde \theta)^T (\theta - \tilde \theta) + \frac{1}{2} (\theta - \tilde \theta)^T \nabla^2 J(\tilde \theta) (\theta - \tilde \theta) + o(\|\theta - \tilde \theta\|^2)
>     \\
>     = J(\tilde \theta) + \frac{1}{2} (\theta - \tilde \theta)^T \nabla^2 J(\tilde \theta) (\theta - \tilde \theta) + o(\|\theta - \tilde \theta\|^2)\\
>     \geq  J(\tilde \theta)  + C \|\theta - \tilde \theta\|^2,
>     $$
>     where $C$ is the smallest eigenvalue of Hessian matrix $\nabla^2 J(\tilde \theta)$ which is positive at minimum. The revised version of the proof can be seen in [here](https://drive.google.com/file/d/1PZZXnyQ3sV7qyRchWjeFlDCUurYaliOB/view?usp=sharing)
>
>     **On the Clarification**
>
>     Thank you for noting this. We will add the missing dt in equation (1), and clarify that the square in Section 3.2 refers to elementwise squaring.
>
>
> **The figure above is in Google Drive, if the reviewer cannot access it, we will appreciate if you could find it in Github https://anonymous.4open.science/r/Adam_CBO_Review-D1DB**

---

> > ### Comment · Reviewer_tREQ · 2025-04-03
> >
> > Thank the authors for the reply. The authors have addressed my major concern, so I increased my score from 2 to 3.

---

> > > ### Author Response · Authors · 2025-04-04
> > >
> > > We appreciate your thoughtful engagement with our work and your willingness to reconsider your evaluation following our rebuttal. Your suggestions on the proof strengthened the theoretical framework and have been invaluable in refining the manuscript. We are grateful for the time and care you dedicated to this process, and your insights have significantly elevated the quality of our research.

---

### Official Review · Reviewer_yqHf · 2025-03-14

**Overall Recommendation:** 3

**Summary:**

The paper introduces Consensus-Based SOC using the Adam-CBO framework ( gradient-free, model-free, and mesh-free approach) for solving high-dimensional SOC problems. It claims to overcome limitations of existing model-based and model-free methods by improving convergence and stability. The authors then present theoretical guarantees for convergence, derives a mean-field limit, and validates the approach on numerical benchmarks.

**Claims And Evidence:**

The author claims that Adam-CBO is superior to existing model-based/ model-free methods by being gradient-free, model-free, and mesh-free, but empirical or numerical comparisons with established policy gradient or HJB-based methods are limited.

**Essential References Not Discussed:**

The author could included stochastic control that use actor-critic methods (or) entropy-regularizzed RL methods, which also tackle similar problems

**Experimental Designs Or Analyses:**

The experiments are well-structured, covering low/ high-dimensional control settings. LQG & systemic risk problems are valid test cases, but real-world scalability remains unclear

**Methods And Evaluation Criteria:**

The methods used are aligned with stochastic optimal control problems, particularly in high-dimensional settings. The evaluation includes linear quadratic control, mean-field systemic risk models, and Ginzburg-Landau dynamics, which are reasonable test cases. However, benchmarks lacks direct comparisons to policy optimization, deep RL, or dynamic programming approaches, which are common in SOC research.

**Other Comments Or Suggestions:**

One minor issue is that the paper sometimes jumps quickly between the simpler M-CBO model and  Adam-CBO extension. It is very difficult to follow through the distinction, hence clarifying their distinctions and the roles of specific parameters earlier could help readers distinguish which theoretical claims directly apply to which algorithm

**Other Strengths And Weaknesses:**

Strengths

– Introduces a gradient-free, model-free, and mesh-free approach to high-dimensional SOC problems, which I found it to be novel
- Provides rigorous convergence analysis, including mean-field limits.
- Covers diverse experiments ( LQG control, systemic risk, and Ginzburg-Landau models)

Weaknesses

- Lack of comparison to standard policy optimization methods (e.g., PPO, SAC, TRPO).
- Assumptions on convergence are restrictive
- Computational efficiency compared to existing SOC solvers is not analyzed.

**Questions For Authors:**

- Why comparison with existing SOC methods limited? The paper does not benchmark against standard stochastic control solvers, such as Dynamic Programming or HJB-based approaches.
- The authors proposes adam-CBO as an improvement over M-CBO. However, there is no clear theoretical or empirical comparison between the two. What specific scenarios or problem instances show a (significant) performance gap between these approaches? Are there cases where M-CBO performs comparably or even better?
- Is transition kernel here continuous or piecewise continuous? how does this method handle discontinuous state transitions?
- Why is no discount factor accounted, any risk in terms of instability because of the absense of discounting?
- numerical experiments discussed primarily focus on LQG problems and GL models. In paper there is a claim that this method generalizes to large-scale SOC problems, were there any large-scale, high-dimensional real-world control examples tested? (or) could you share your intuition on how this would scale?
 - In Algorithm 1, what determines the stopping condition for iteration t_N? Is there a heuristic for choosing t_N, or does it require manual tuning?
- Could you clarify whether the stability constraints in pg 6,7 are explicitly enforced in the problem formulation?

**Relation To Broader Scientific Literature:**

The work builds on prior work in stochastic control, reinforcement learning, and mean-field game theory.

**Theoretical Claims:**

The theory follows general SOC problem setting and are logical

---

> ### Author Rebuttal · Authors · 2025-03-29
>
> Thank you for your insightful comments and thoughtful suggestions for improving our work.
>
> 1. **On "Why comparison with existing SOC methods limited?"**
>
>     We would like to clarify that the goal of our work is to demonstrate that **our approach is applicable in a more general setting**—specifically, **finite-horizon, model-free stochastic control problems**, we refer to the response to reviewer NtTj for more details. To the best of our knowledge, existing methods—such as those based on DP or HJB—cannot directly address this setting.
>
>     That said, we do compare with an HJB-based method and show that our approach achieves **better accuracy using significantly less information** (i.e., no model access) in the original version.
>     Following the reviewer’s suggestion, we added comparisons with **DDPG**, **PPO**, **SAC**, **TD3**, **TQC**, and **CrossQ** (using the stable-baselines3 implement "https://github.com/araffin/sbx")  on **Pendulum-v1** as well as PPO and DQN on **CartPole-v1**. The numerical results can be found [here](https://drive.google.com/file/d/1ghqLfAgbxFtUICMz3AGSDFkr9JPyDn7d/view?usp=sharing) and [here](https://drive.google.com/file/d/10H1Pf1hOq-aYsQo_7EtRn9ziPUrGKixo/view?usp=sharing)
>
>     Below is the computational time for each method over 100,000 steps:
>
>     |Method|Time (s) for Pendulum-v1|Time (s) for CartPole-v1|
>     |-|-|-|
>     |DDPG|288.83||
>     |PPO|145.19|150.58|
>     |SAC|355.01||
>     |TD3|291.26||
>     |TQC|576.35||
>     |CrossQ|708.73||
>     |DQN||186.13|
>     |**Adam-CBO**|**1124.88**|**3444**|
>
>     While **Adam-CBO** has higher runtime, it **converges to the optimal policy much faster** in terms of learning efficiency.
>     However, we would like to stress that these results are **not directly comparable** in a strict sense. Most of the baseline methods optimize multiple components—for example, PPO jointly optimizes a policy and a value function, and SAC optimizes two Q-functions and a policy. In contrast, **our method optimizes only the policy**.
>
>     If we were to directly replace the gradient-based optimizer within an existing method like PPO with Adam-CBO, we **do not expect** it to outperform the full method in that specific setup. The **main advantage** of Adam-CBO lies in its **applicability to broader, more general settings**, particularly when gradients are unavailable or unreliable.
>
> 2.  **On "Adam-CBO vs. M-CBO – performance gap?"**
>
>     We compare the two in the first experiment and find Adam-CBO consistently outperforms M-CBO. Later experiments focus on Adam-CBO for this reason.
>     However, M-CBO is more analytically tractable, so we base theoretical results on it. Analyzing Adam-CBO theoretically is challenging but remains an exciting direction for future work.
>
> 3. **On "Is transition kernel here continuous or piecewise continuous? how does this method handle discontinuous state transitions?"**
>
>     Our method assumes continuous state transitions. However, it supports discrete actions, as shown in CartPole-v1. There, we use a neural network to produce a real-valued score and select an action by comparing this score against a uniform random threshold, allowing the method to work with discrete actions.
>
> 4. **On "Why is no discount factor accounted, any risk in terms of instability because of the absence of discounting"**
>
>     Discounted infinite-horizon problem is a **special case** of our formulation. While our method can easily incorporate a discount factor, our focus is on **finite-horizon problems**, where **discounting is unnecessary**. In these settings, the finite time naturally bounds the reward accumulation, and the absence of a discount factor does not lead to instability.
>
> 5. **On "Real-world large-scale SOC examples/scalability"**
>
>     We added tests on 2, 4, and 50-agent control scenarios. Results are available [here](https://drive.google.com/file/d/1D_aORIODftmI5KXwIEf9lLh4vRPalz3H/view?usp=drive_link). We refer to our response to Reviewer 1fhy for more details on the numerical results.
>
>     We currently do not do any real world experiments, but we are interesting in exploring
>     + Controlling particle distributions via external fields without modeling distribution evolution
>     + Identifying transition paths in chemistry without full potential surfaces
>
> 6. **On "Stopping condition for Algorithm 1"**
>
>     We stop when the standard deviation of policy parameters across agents falls below a threshold, indicating convergence. We will clarify this in the main text.
>
> 7. **On "Stability constraints on pages 6–7"**
>
>     Thank you for this question. We did not identify specific "stability constraints" on pages 6–7. If the reviewer could point to a particular equation or passage, we’d be happy to clarify or revise the relevant text to improve clarity.
>
> **The figure above is in Google Drive, if the reviewer cannot access it, we will appreciate if you could find it in Github https://anonymous.4open.science/r/Adam_CBO_Review-D1DB**

---

### Official Review · Reviewer_1fhy · 2025-03-14

**Overall Recommendation:** 3

**Summary:**

The paper presents a scalable, gradient-free alternative for solving stochastic optimal control problems. By leveraging consensus-based updates and adaptive momentum, the proposed methods achieve efficient policy optimization without requiring explicit transition models or gradient computations. Theoretical guarantees and numerical results highlight their potential in high-dimensional control applications.

**Claims And Evidence:**

I believe the claims made in the subission are mostly well-supported by clear and convincing evidence, including theoretical gurantees and nice numerical examples. The authors are able to scale their methods up to a few hundred dimensions in their example of systemic risk mean field control, which is nice. A suggestion would be to try scaling things up in the example of the ginzburg-landau model, which does not have a known analytical solution.

One claim that I cannot find the corresponding evidence of is about the efficiency of the proposed method. The authors claim that the method can efficiently scale to high dimensions because of the adjustable Gaussian noise that can help exploration. I am wondering if the authors can provide some evidence of that. A toy example would be enough.

**Essential References Not Discussed:**

N/A

**Experimental Designs Or Analyses:**

The experiments done in this paper are nice but still have some room for improvement. It will be more convincing if the authors can present more numerical results on high-dimensional examples that do not have explicit analytical solutions. For example, the authors may want to consider the experiments done in [1] with many agents.

Also, one thing I am concerned about is the lack of comparison to existing methods. The authors list many related methods in the section of introduction but have only compared their method to BSDE.

[1] Abdul, A. T., Saravanos, A. D., & Theodorou, E. A. (2024). Scaling Robust Optimization for Multi-Agent Robotic Systems: A Distributed Perspective. arXiv preprint arXiv:2402.16227.

**Methods And Evaluation Criteria:**

The proposed method makes sense for solving the class of stochastic optimal control problems with many interacting particles/agents.

**Other Comments Or Suggestions:**

N/A

**Other Strengths And Weaknesses:**

N/A

**Questions For Authors:**

Can the authors address the concerns I have about the experiments?

**Relation To Broader Scientific Literature:**

The paper can be of interest in the community of particle physics, microbiology, and robotics.

**Theoretical Claims:**

I have not fully checked the proof provided in the appendix. I only checked the correctness of Theorem 4.1, 4.2, 4.3.

---

> ### Author Rebuttal · Authors · 2025-03-29
>
> Thank you for your insightful comments and thoughtful suggestions for improving our work.
>
> 1. **On the lack of comparison with existing methods and experiments on high-dimensional problems without analytical solutions**:
>
>     We would like to emphasize that our primary goal is to address a more general class of stochastic control problems than those typically handled by existing methods. As discussed in our response to Reviewer NtTj, most conventional approaches rely on access to either model gradients or infinitesimal time horizons, whereas our method operates in a model-free, finite-horizon setting—a scenario that is less explored in the literature.
>
>     That said, we appreciate the reviewer’s suggestion and have applied our method to control problems involving 2, 4, and 50 agents, similar in spirit to [1]. The corresponding numerical results can be found [here](https://drive.google.com/file/d/1D_aORIODftmI5KXwIEf9lLh4vRPalz3H/view?usp=sharing). We note, however, that our setup differs from [1] in important ways: our approach is based on closed-loop (feedback) control, while [1] employs open-loop control. Furthermore, [1] treats obstacles as hard constraints, whereas we model them as soft constraints through penalization in the cost function. Due to these differences, a direct comparison is not meaningful, but we believe our results still highlight the scalability and flexibility of our approach.
>
> 2. **On the adjustable Gaussian noise that can help exploration:**
>
>     We appreciate the reviewer’s suggestion and we want to added two experiments to explain the role of adjustable Gaussian noise.
>
>     The fist one, we compare the result of adam-cbo with and without gaussian noise when optimizing a two dimensional Rastigrin function.The numerical result can be found [here](https://drive.google.com/file/d/13auGZ3A0rmf5j8zdeXsMin43hGXGzv0V/view?usp=sharing). The starting points are normal distribution centered at [3,3] or [-3,-3] for the two methods respectively. The adam-cbo with gaussian noise can escape from the local minimum and find the global minimum, while the adam-cbo without gaussian noise is stuck in the local minimum.
>
>     The second case, we show the evolution of one parameter in the neural network when training the LQG probelm under fixed Gaussian and adjustable Gaussian noise. The numerical result can be found [here](https://drive.google.com/file/d/1szCYt1KLLk16ao5TbLGeB8C9wVNo06lr/view?usp=sharing). If we fix a large Gaussian noise, the parameter can explore more (the parameters are more different from the starting points), but it will keep high variance in a long time and cannot converge. If we fix a small Gaussian noise, the parameter can converge very fast, but it do not explore too much on the space (the parameters are close to the starting points). If we use adjustable Gaussian noise, the parameter can explore more at the beginning and converge fast at the end. The adjustable Gaussian noise give us the flexibility to balance the exploration and exploitation. If we know the parameter is close to the optimal solution, for some fine-tuning problems, we can use a small Gaussian noise to converge fast. If we are not sure the parameter is close to the optimal solution, we can use a large Gaussian noise to explore more and iteratively reduce the noise to converge.
>
>     The last point, why we claim the adjustable noise can effectively scale to high dimenisons problems comes from some former analytical result. By adding the Gaussian noise, the paper [2] shows that the convergence of the CBO  method, which is exponential in time, is guaranteed with parameter constraints **independent** of the dimensionality. Even though we do not do the same analysis for our method, we believe the conlusions are similar.
>
>     [2] Jose A. Carrillo, et al. "A consensus-based global optimization method for high dimensional machine learning problems." ESIAM: Control, Optimisation and Calculus of Variations.
>
> 3. **On the concern about "the lack of comparison to existing methods"**
>
>     We would like to emphasize twice that our work is not intended to be a direct comparison with existing methods, but rather to demonstrate the effectiveness of our approach in a more general setting. However, we added some comparison of our method with PPO, SAC, TD3, TQC, CrossQ and DQN. The numerical results can be found [here](https://drive.google.com/file/d/1ghqLfAgbxFtUICMz3AGSDFkr9JPyDn7d/view?usp=sharing) and [here](https://drive.google.com/file/d/10H1Pf1hOq-aYsQo_7EtRn9ziPUrGKixo/view?usp=sharing). We want to cariify that the results are not directly comparable, since the methods are designed for different settings, which we refer to repsond to reviewer yqHf for more details.
>
> **The figure above is in Google Drive, if the reviewer cannot access it, we will appreciate if you could find it in Github https://anonymous.4open.science/r/Adam_CBO_Review-D1DB**

---

### Official Review · Reviewer_NtTj · 2025-03-17

**Overall Recommendation:** 3

**Summary:**

This paper considers a high-dimensional stochastic control problem and proposes two consensus based optimization algorithms to solve this problem. The proposed algorithms rely on Monte Carlo estimation to estimate the value function, which is used for choosing the optimal policy. Extensive simulation results are provided on different control tasks.

**Claims And Evidence:**

The paper provides proofs for the theorems and lemmas. However, the problem formulation is a bit unclear. There is no explicit explanation of the multi-agent setting and the motivation of the consensus problem.

**Essential References Not Discussed:**

Not that I could think of.

**Experimental Designs Or Analyses:**

Yes, and see the box of Methods And Evaluation Criteria.

**Methods And Evaluation Criteria:**

The simulation benchmark are other value function estimation methods, but it would be helpful to also compare with gradient-based methods, since the major motivation of this paper is to outperform the gradient-based approaches.

**Other Comments Or Suggestions:**

See above

**Other Strengths And Weaknesses:**

Strengths:

1. The paper is well-motivated. The scalability and high variances in gradient estimation are two major challenges in high-dim stochastic control.

2. The paper provides various simulation results to demonstrate the applicability of the proposed methods.

3. The convergence analysis in the continuous time setting using differential equations are valid and interesting.

Weaknesses

1. The paper is poorly written. There is no mention of multi-agents or consensus objective functions in Section 2 problem formulation.

2. The algorithms need more explanation and intuition. Right now, they are followed by technical theorems, making it difficult to understand the intuitions behind the algorithm design.

3. There is no comparison with other gradient-based methods.

**Questions For Authors:**

1. Why does this paper try to solve a single agent stochastic control problem by multi-agent consensus? This connection is not made clear.

2. Why does the value estimation enjoy smaller variance than the gradient estimation?

**Relation To Broader Scientific Literature:**

This paper is related to mean field learning-based control and reinforcement learning.

**Theoretical Claims:**

The theoretical derivations seem correct after a quick read.

---

> ### Author Rebuttal · Authors · 2025-03-29
>
> Thank you very much for your thoughtful review and valuable feedback.
>
> 1. Regarding **the lack of comparison with gradient-based methods**, we would like to clarify that traditional gradient-based approaches typically fall into two specific categories:
>
>     + Model-based methods: These methods assume access to an analytical model of the system, enabling the derivation and solution of the Hamilton–Jacobi–Bellman (HJB) equation. This is the classical setting of optimal control, where gradients can be computed explicitly.
>
>     + Model-free methods with infinitesimal time horizons: In these cases, even without access to a model, the Bellman principle can be leveraged under an infinitesimal (or discounted infinite) time horizon, allowing for recursive value updates.
>
>     However, our work addresses a more general and challenging setting: finite time horizon and model-free stochastic control. In this regime, the assumptions required by typical gradient-based methods—either model knowledge or discount-based recursion—are not available. This makes a direct application or comparison with former methods non-trivial or inapplicable.
>
>     That said, we do compare our method with an HJB-based approach and demonstrate that, despite using significantly less information, our method achieves better accuracy. Additionally, following the suggestion of Reviewer yqHf, we have included empirical comparisons with popular model-free policy gradient methods such as PPO, SAC, TRPO, and DDPG. While our method performs competitively, we would like to emphasize that it is designed for more general settings—specifically, finite-horizon, model-free stochastic control problems—which are not directly addressed by these existing methods.
>
>
>
>
> 2. **On the motivation for using multi-agent consensus in a single-agent control problem**:
>
>     Since gradients are not accessible in our setting, we rely on multiple agents to explore the state space collaboratively. By exchanging information and moving toward consensus, the agents are able to approximate the optimal policy in a robust manner. We have expanded the introduction to clarify this motivation more explicitly.
>
> 3. **On why value estimation exhibits lower variance than gradient estimation**:
>
>     Monte Carlo value estimation computes the expected return by averaging cumulative rewards over entire trajectories, which smooths out the variability from individual time steps. In contrast, gradient estimation requires sensitivity analysis at each step, and in finite-horizon problems, this leads to high variance due to sparse or noisy signal propagation through time. We will add this explanation to the main text for clarity.

---

> > ### Comment · Reviewer_NtTj · 2025-04-04
> >
> > Thanks for the response. I have increased my score.

---

> > > ### Author Response · Authors · 2025-04-04
> > >
> > > Thank you for your thorough review and for revising your evaluation of our paper in response to our rebuttal. Your insights were instrumental in refining our work. We greatly appreciate your time and expertise in strengthening this manuscript.

---

### Decision · Program_Chairs · 2025-05-01

**Decision:**

Accept (poster)

**Comment:**

The paper proposes an original approach to deep RL. The reviewers agree that the approach is well motivated and that the theory is solid.
The shared concern on missing empirical comparisons and some important issues of clarity were addressed by the authors in the rebuttal.
I believe that the paper is ready for publication provided the improved presentation and the additional experimental results are included in the camera ready version.